# Continual Nonlinear ICA-based Representation Learning

## Abstract

Unsupervised identification of disentangled representations remains a challenging problem. Recent progress in nonlinear Independent Component Analysis (ICA) provides a promising causal representation learning framework by separating latent sources from observable nonlinear mixtures. However, its identifiability hinges on the incorporation of side information, such as time or domain indexes, which are challenging to obtain adequately offline in real-world scenarios. In this paper, we develop a novel approach for nonlinear ICA that effectively accommodates continually arriving domains. We first theoretically demonstrate that model identifiability escalates from subspace to component-wise level with the increment of domains. It motivates us to maintain prior knowledge and progressively refine it using new arriving domains. Upon observing a new domain, our approach optimizes the model by satisfying two objectives: (1) reconstructing the observations within the current domain, and (2) preserving the reconstruction capabilities for prior domains through gradient constraints. Experiments demonstrate that our method achieves performance comparable to nonlinear ICA methods trained jointly on multiple offline domains, demonstrating its practical applicability in continual learning scenarios.

## 1 Introduction

Causal representation learning aims at recovering high-level semantic variables from low-level observations and their casual relations. Compared with current deep learning models which are trained as black-box functions, it is more explainable and generalizable by the identification of the underlying causal generation process. A well-established theoretical result demonstrates that reconstructing these latent variables in a fully unsupervised way is impossible without further assumptions (Hyvärinen & Pajunen, 1999). Among multiple efforts towards this problem (Xie et al., 2020; Silva et al., 2006), nonlinear ICA attracts a lot of attention by providing a promising framework and demonstrating an identification guarantee.

Nonlinear ICA focuses on recovering independent latent variables from their nonlinear mixtures. Denote an observed $n$-dimensional vector by $\mathbf{x}$, which is generated by a number of independent latent variables $\mathbf{z}$ through an arbitrary invertible mixing function $g$ as $\mathbf{x} = g(\mathbf{z})$. The objective of nonlinear ICA is to reconstruct the latent variables $\mathbf{z}$ by discovering the inverse function $g^{-1}$ based on the observation $\mathbf{x}$ only in an unsupervised manner. Apparently, without additional constraints, we can never find out a meaningful solution. More rigorously, the identifiability of nonlinear ICA cannot be guaranteed when only relying on independence assumption (Hyvärinen & Pajunen, 1999).

To address this problem, existing works focus on adding constraints on the mixing function (Gresele et al., 2021; Buchholz et al., 2022; Zheng et al., 2022), or most popularly, benefiting from the non-stationary of source data (Hyvarinen & Morioka, 2016; 2017; Hyvarinen et al., 2018; Khemakhem et al., 2019) to advance the identifiability. By introducing auxiliary variable $\mathbf{u}$ and assuming the non-i.i.d sources are conditionally independent given $\mathbf{u}$, the latent variables can be estimated up to component-wise identifiable. Although current research on nonlinear ICA has made great progress, it still relies on observing sufficient domains simultaneously, which limits its application to scenarios where changing domains may arrive sequentially. Specifically, the model trained with sequential arrival of domains without making adjustments is equivalent to the scenario where only one domain is observed. Consequently, the model becomes unidentifiable.

In this paper, we present a novel approach to learning causal representation in continually arriving domains. Distinct from traditional continual classification tasks, continual causal representation learning (CCRL) requires that the model leverages the changes in distribution across varying domains. This implies that the problem cannot be segregated into discrete local learning tasks, such as learning causal representation within individual domains and subsequently fusing them. In this context, we conduct a theoretical examination of the relationships between model identification and the number of observed domains. Our research indicates that the identifiability increases with the inclusion of additional domains. In particular, subspace identification can be achieved with $n + 1$ domains, while component-wise identification necessitates $2n + 1$ domains or more. This indicates that when the domain count is inadequate ($n + 1$), we can only identify the manifold spanned by a subset of latent variables. However, by utilizing the new side information in the distribution change of arriving domains, we can further disentangle this subset.

This discovery motivates us to develop a method that retains prior knowledge and refines it using information derived from incoming domains, a process reminiscent of human learning mechanisms. To realize causal representation learning, we employ two objectives: (1) the reconstruction of observations within the current domain, and (2) the preservation of reconstruction capabilities for preceding domains via gradient constraints. To accomplish these goals, we apply Gradient Episodic Memory (GEM) (Lopez-Paz & Ranzato, 2017) to constrain the model's gradients. GEM aligns the gradients of the new domain with those of prior domains by eliminating factors within the current domain that are detrimental to previous domains. Through empirical evaluations, we demonstrate that our continual approach delivers performance on par with nonlinear ICA techniques trained jointly across multiple offline domains. Importantly, the guarantee of identifiability persists even when incoming domains do not introduce substantial changes for partial variables. Furthermore, we demonstrate that the sequential order of domains can maintain the identification process of partial variables in causal representation learning.

## 2 RELATED WORK

**Causal representation learning.** Beyond conventional representation learning, causal representation learning aims to identify the underlying causal generation process and recover the latent causal variables. There are pieces of work aiming towards this goal. For example, it has been demonstrated in previous studies that latent variables can be identified in linear-Gaussian models by utilizing the vanishing Tetrad conditions (Spearman, 1928), as well as the more general concept of t-separation (Silva et al., 2006). Additionally, the Generalized Independent Noise (GIN) condition tried to identify a linear non-Gaussian causal graph (Xie et al., 2020). However, all of these methods are constrained to the linear case while nonlinear ICA provides a promising framework that learns identifiable latent causal representations based on their non-linear mixture. However, the identifiability of nonlinear ICA has proven to be a challenging task (Hyvärinen & Pajunen, 1999), which always requires further assumptions as auxiliary information, such as temporal structures (Sprekeler et al., 2014), non-stationarities (Hyvarinen & Morioka, 2016; 2017), or a general form as auxiliary variable (Hyvarinen et al., 2018). These methods indicate that sufficient domains (changes) are crucial for ensuring the identifiability of nonlinear ICA. In this paper, we consider the scenario that changing domains may arrive not simultaneously but sequentially or even not adequately.

**Continual learning.** In conventional machine learning tasks, the model is trained on a dedicated dataset for a specific task, then tested on a hold-out dataset drawn from the same distribution. However, this assumption may contradict some real-world scenarios, where the data distribution varies over time. It motivates researchers to explore continual learning to enable an artificial intelligence system to learn continuously over time from a stream of data, tasks, or experiences without losing its proficiency in the ones it has already learned. The most common setting is class incremental recognition (Rebuffi et al., 2017; Hou et al., 2019; Van De Ven et al., 2021), where new unseen classification categories with different domains arrive sequentially. To solve this problem, existing methods are commonly divided into three categories. Regulization-based methods (Riemer et al., 2018; Zeng et al., 2019; Farajtabar et al., 2020; Saha et al., 2021; Tang et al., 2021; Wang et al., 2021) add the constraints on the task-wise gradients to prevent the catastrophic forgetting when updating network weights for new arriving domains. Memory-based methods (Robins, 1995; Rebuffi et al., 2017; Lopez-Paz & Ranzato, 2017; Chaudhry et al., 2018; 2019; Hu et al., 2019; Kemker & Kanan, 2017; Shin et al., 2017; Pellegrini et al., 2020; Van De Ven et al., 2021) propose to store previous knowledge

in a memory, such as a small set of examples, a part of weights, or episodic gradients to alleviate forgetting. Distillation-based methods (Li & Hoiem, 2017; Rebuffi et al., 2017; Hou et al., 2019; Castro et al., 2018; Wu et al., 2019; Yu et al., 2020; Tao et al., 2020; Liu et al., 2020; Mittal et al., 2021) remember the knowledge trained on previous tasks by applying knowledge distillation between previous network and currently trained network. Please note that CCRL is distinct from conventional class incremental recognition. It is because CCRL needs to leverage the domain change (comparing two domains) to identify the latent variables. This implies that the problem cannot be divided into discrete local learning tasks, such as learning causal representation within individual domains and then merging them together, while training separate networks for different tasks will definitely reach state-of-the-art performance in a continual classification learning scenario. Thus, we introduce a memory model to store the information of previous domains and use it to adjust the model parameters.

# 3 IDENTIFIABLE NONLINEAR ICA WITH SEQUENTIALLY ARRIVING DOMAINS

In this section, we conduct a theoretical examination of the relationship between model identification and the number of domains. Initially, we introduce the causal generation process of our model (in Section 3.1), which considers the dynamics of changing domains. Subsequently, we demonstrate that model identifiability improves with the inclusion of additional domains. More specifically, we can achieve component-wise identification with $2n + 1$ domains (in Section 3.2.1), and subspace identification with $n + 1$ domains (in Section 3.2.2). Building on these theoretical insights, we introduce our method for learning causal representation in the context of continually emerging domains (in Section 3.3).

## 3.1 PROBLEM SETTING

As shown in Figure 1, we consider the data generation process as follows:

$$\mathbf{z}_c \sim p_{\mathbf{z}_c}, \quad \tilde{\mathbf{z}}_s \sim p_{\tilde{\mathbf{z}}_s}, \quad \mathbf{z}_s = f_\mathbf{u}(\tilde{\mathbf{z}}_s), \quad \mathbf{x} = g(\mathbf{z}_c, \mathbf{z}_s), \tag{1}$$

where $\mathbf{x} \in \mathcal{X} \subseteq \mathbb{R}^n$ are the observations mixed by latent variables $\mathbf{z} \in \mathcal{Z} \subseteq \mathbb{R}^n$ through an invertible and smooth nonlinear function $\mathbf{g} : \mathcal{Z} \to \mathcal{X}$. The latent variables $\mathbf{z}$ can be partitioned into two groups: changing variables $\mathbf{z}_s \in \mathcal{Z}_s \subseteq \mathbb{R}^{n_s}$ whose distribution changes across domains $\mathbf{u}$, and invariant variables $\mathbf{z}_c \in \mathcal{Z}_c \subseteq \mathbb{R}^{n_c}$ which remains invariant. Given $T$ domains in total, we have $p_{\mathbf{z}_s|\mathbf{u}_k} \neq p_{\mathbf{z}_s|\mathbf{u}_l}, p_{\mathbf{z}_c|\mathbf{u}_k} = p_{\mathbf{z}_s|\mathbf{u}_l}$ for all $k, l \in \{1, \ldots, T\}, k \neq l$. We parameterize the influence of domains $\mathbf{u}$ for changing variables $\mathbf{z}_s$ as the function of $\mathbf{u}$ to its parent variables $\tilde{\mathbf{z}}_s$, i.e. $\mathbf{z}_s = f_\mathbf{u}(\tilde{\mathbf{z}}_s)$. One can understand this setting with the following example: suppose the higher level variables follow Gaussian distribution, i.e., $\tilde{\mathbf{z}}_s \sim \mathcal{N}(\mathbf{0}, \mathbf{I})$, and $\mathbf{u}$ could be a vector denoting the variance of the distribution. The combination of $\mathbf{u}$ with $\tilde{\mathbf{z}}_s$ will produce a Gaussian variable with different variances at different domains. In this paper, we assume $\tilde{\mathbf{z}}_s$ follows the Gaussian distribution to make it tractable.

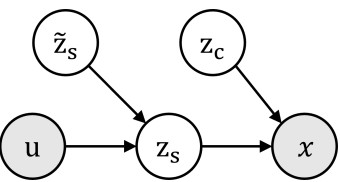

Figure 1: **Data generation process.** $\mathbf{x}$ is influenced by variables $\mathbf{z}_s$ (change with different domains $\mathbf{u}$) and invariant variables $\mathbf{z}_c$.

The objective of nonlinear ICA is to recover the latent variables $\mathbf{z}_s$ and $\mathbf{z}_c$ given the observation $\mathbf{x}$ and domain variables $\mathbf{u}$ by estimating the unmixing function $\mathbf{g}^{-1}$. In this paper, we consider the case where domains arrive sequentially, i.e., we aim to recover the latent variables by sequentially observing $\mathbf{x}|\mathbf{u}_1, \mathbf{x}|\mathbf{u}_2, \ldots, \mathbf{x}|\mathbf{u}_T$.

## 3.2 IDENTIFIABILITY THEORY OF NONLINEAR ICA

The identifiability is the key to nonlinear ICA to guarantee meaningful recovery of the latent variables. Mathematically, the identifiability of a model is defined as

$$\forall (\boldsymbol{\theta}, \boldsymbol{\theta}') : \quad p_{\boldsymbol{\theta}}(\mathbf{x}) = p_{\boldsymbol{\theta}'}(\mathbf{x}) \implies \boldsymbol{\theta} = \boldsymbol{\theta}', \tag{2}$$

where $\boldsymbol{\theta}$ represents the parameter generating the observation $\mathbf{x}$. That is, if any two different choices of model parameter $\boldsymbol{\theta}$ and $\boldsymbol{\theta}'$ lead to the same distribution, then this implies that $\boldsymbol{\theta}$ and $\boldsymbol{\theta}'$ are equal

(Khemakhem et al., 2019). For our data generation defined in equation 1, we have $\boldsymbol{\theta} = (g, \mathbf{z}_c, \mathbf{z}_s)$, and $\boldsymbol{\theta}' = (\hat{g}, \hat{\mathbf{z}}_c, \hat{\mathbf{z}}_s)$ which denotes the estimated mixing function, estimated invariant variables, and estimated changing variables respectively. Thus, a fully identifiable nonlinear ICA needs to satisfy at least two requirements: the ability to reconstruct the observation and the complete consistency with the true generating process. Unfortunately, current research cannot achieve this level of identifiability without further assumptions that are considerably restrictive. Therefore, existing works typically adopt a weaker notion of identifiability. In the following, we discuss two types of identifiability for the changing variable, and show that the identifiability progressively increases from subspace identifiability to component-wise one by incorporating more domains.

In this work, we follow (Kong et al., 2022) and assume our estimated latent process $(\hat{g}, \hat{\mathbf{z}}_c, \hat{\mathbf{z}}_s)$ could generate observation $\hat{\mathbf{x}}$ with identical distribution with observation $\mathbf{x}$ generated by the true latent process $(g, \mathbf{z}_c, \mathbf{z}_s)$, i.e.,

$$p_{\mathbf{x}|\mathbf{u}}(\mathbf{x}'|\mathbf{u}') = p_{\hat{\mathbf{x}}|\mathbf{u}}(\mathbf{x}'|\mathbf{u}'), \quad \mathbf{x}' \in \mathcal{X}, \mathbf{u}' \in \mathcal{U}. \tag{3}$$

### 3.2.1 Component-wise Identifiability for Changing Variable

First, we show that the changing variable can be identified up to permutation and component-wise invertible transformation with sufficient changing domains. Specifically, for the true latent changing variable $\mathbf{z}_s$, there exists an invertible function $h = g^{-1} \circ \hat{g} : \mathbb{R}^{n_s} \to \mathbb{R}^{n_s}$ such that $\mathbf{z}_s = h(\hat{\mathbf{z}}_s)$, where $h$ is composed of a permutation transformation $\pi$ and a component-wise nonlinear invertible transformation $A$, i.e., $\hat{g} = g \circ \pi \circ A$ [1]. That is, the estimated variable $\hat{z}_j$ and the true variable $z_i$ have a one-to-one correspondence with an invertible transformation for $\forall i, j \in \{1, \ldots, n_s\}$. We have the following lemma from (Kong et al., 2022).

**Lemma 1** *Suppose that the data generation process follows equation 1 and that the following assumptions hold:*

1. *The set $\{\mathbf{z} \in \mathbb{Z} \mid p(\mathbf{z}) = 0\}$ has measure zero.*

2. *The probability density given each domain should be sufficiently smooth. i.e., $p_{\mathbf{z}|\mathbf{u}}$ is at least second-order differentiable.*

3. *Given domain $\mathbf{u}$, every element of latent variable $\mathbf{z}$ should be independent with each other. i.e., $z_i \perp\!\!\!\perp z_j | \mathbf{u}$ for $i, j \in \{1, \ldots, n\}$ and $i \neq j$.*

4. *For any $\mathbf{z}_s \in \mathcal{Z}_s$, there exists $2n_s + 1$ values of $\mathbf{u}$, such that for $k = 1, \ldots, 2n_s$, $i = 1, \ldots, n_s$, the following matrix is invertible:*

$$\begin{bmatrix} \phi_1''(\mathbf{1},\mathbf{0}) & \ldots & \phi_i''(\mathbf{1},\mathbf{0}) & \ldots & \phi_{n_s}''(\mathbf{1},\mathbf{0}) & \phi_1'(\mathbf{1},\mathbf{0}) & \ldots & \phi_i'(\mathbf{1},\mathbf{0}) & \ldots & \phi_{n_s}'(\mathbf{1},\mathbf{0}) \\ \vdots & \ddots & \vdots & \vdots & \vdots & \vdots & \vdots & \ddots & \vdots & \vdots \\ \phi_1''(\mathbf{k},\mathbf{0}) & \ldots & \phi_i''(\mathbf{k},\mathbf{0}) & \ldots & \phi_{n_s}''(\mathbf{k},\mathbf{0}) & \phi_1'(\mathbf{k},\mathbf{0}) & \ldots & \phi_i'(\mathbf{k},\mathbf{0}) & \ldots & \phi_{n_s}'(\mathbf{k},\mathbf{0}) \\ \vdots & \ddots & \vdots & \vdots & \vdots & \vdots & \vdots & \ddots & \vdots & \vdots \\ \phi_1''(\mathbf{2n_s},\mathbf{0}) & \ldots & \phi_i''(\mathbf{2n_s},\mathbf{0}) & \ldots & \phi_{n_s}''(\mathbf{2n_s},\mathbf{0}) & \phi_1'(\mathbf{2n_s},\mathbf{0}) & \ldots & \phi_i'(\mathbf{2n_s},\mathbf{0}) & \ldots & \phi_{n_s}'(\mathbf{2n_s},\mathbf{0}) \end{bmatrix},$$

*where*

$$\phi_i''(\mathbf{k},\mathbf{0}) := \frac{\partial^2 \log(p_{\mathbf{z}|\mathbf{u}}(z_i|\mathbf{u_k}))}{\partial z_i^2} - \frac{\partial^2 \log(p_{\mathbf{z}|\mathbf{u}}(z_i|\mathbf{u_0}))}{\partial z_i^2}, \phi_i'(\mathbf{k},\mathbf{0}) := \frac{\partial \log(p_{\mathbf{z}|\mathbf{u}}(z_i|\mathbf{u_k}))}{\partial z_i} - \frac{\partial \log(p_{\mathbf{z}|\mathbf{u}}(z_i|\mathbf{u_0}))}{\partial z_i}$$

*are defined as as the difference between second-order derivative and first-order derivative of log density of $z_i$ between domain $\mathbf{u_k}$ and domain $\mathbf{u_0}$ respectively,*

*Then, by learning the estimation $\hat{g}, \hat{\mathbf{z}}_c, \hat{\mathbf{z}}_s$ to achieve equation 3, $\mathbf{z}_s$ is component-wise identifiable.* [2]

The proof can be found in Appendix A1.2. Basically, the theorem states that if the distribution of latent variables is "complex" enough and each domain brings enough changes to those changing variables, those changing variables $\mathbf{z}_s$ are component-wise identifiable.

---

[1] More formally "component-wise nonlinear identifiability" as it doesn't require exactly identify each element.

[2] We only focus on changing variables $\mathbf{z}_s$ in this paper. One may refer (Kong et al., 2022) for those who are interested in the identifiability of $\mathbf{z}_c$.

**Repeated distributions of partially changing variables.** Previous works assume that when the domain changes, the changing variables will undergo a distribution shift. However, this assumption may be overly restrictive for practical scenarios, because there is no guarantee or clear justification that the data distribution of all changing variables will change when the domain changes. Specifically, there may exist domains with the same distribution for partial variables:

$$p_{\mathbf{z}|u}(z_i|\mathbf{u_k}) = p_{\mathbf{z}|u}(z_i|\mathbf{u_l}) \quad \exists k, l \in \{0, \ldots, 2n_s\}, k \neq l, i \in \{1, \ldots, n_s\}. \tag{4}$$

In practical human experience, we frequently encounter novel information that enhances or modifies our existing knowledge base. Often, these updates only alter specific aspects of our understanding, leaving the remainder intact. This raises an intriguing question about model identifiability: Does such partial knowledge alteration impact the invertibility of the matrix, as delineated in Assumption 4 of Lemma 1? In response to this issue, we present the following remark, with further details provided in Appendix A1.3.

**Remark 1** *For $n_s \geq 2$ and we use $|S_i|$ to denote the cardinality of non-repetitive distributions of latent changing variable $z_i$ $(1 \leq |S_i| \leq T)$. If Lemma 1 hold, then $|S_i| \geq 3$ for every $i \in \{1, \ldots, n_s\}$.*

### 3.2.2 SUBSPACE IDENTIFIABILITY FOR CHANGING VARIABLE

Although component-wise identifiability is powerful and attractive, holding $2n_s + 1$ different domains with sufficient changes remains a rather strong condition and may be hard to meet in practice. In this regard, we investigate the problem of what will happen if we have fewer domains. We first introduce a notion of identifiability that is weaker compared to the component-wise identifiability discussed in the previous section.

**Definition 1 (Subspace Identifiability of Changing Variable)** *We say that the true changing variables $\mathbf{z}_s$ are subspace identifiable if, for the estimated changing variables $\hat{\mathbf{z}}_s$ and each changing variable $z_{s,i}$, there exists a function $h_i : \mathbb{R}^{n_s} \to \mathbb{R}$ such that $z_{s,i} = h_i(\hat{\mathbf{z}}_s)$.*

We now provide the following identifiability result that uses a considerably weaker condition (compared to Lemma 1) to achieve the subspace identifiability defined above, using only $n_s + 1$ domains.

**Theorem 1** *Suppose that the data generation process follows equation 1 and that Assumptions 1, 2, and 3 of Lemma 1 hold. For any $\mathbf{z}_s \in \mathcal{Z}_s$, we further assume that there exists $n_s + 1$ values of $\mathbf{u}$ such that for $i = 1, \ldots, n_s$ and $k = 1, \ldots, n_s$, the following matrix*

$$\begin{bmatrix} \phi_1'(\mathbf{1}, \mathbf{0}) & \ldots & \phi_i'(\mathbf{1}, \mathbf{0}) & \ldots & \phi_{n_s}'(\mathbf{1}, \mathbf{0}) \\ \vdots & \ddots & \vdots & \vdots & \vdots \\ \phi_1'(\mathbf{k}, \mathbf{0}) & \ldots & \phi_i'(\mathbf{k}, \mathbf{0}) & \ldots & \phi_{n_s}'(\mathbf{k}, \mathbf{0}) \\ \vdots & \vdots & \vdots & \ddots & \vdots \\ \phi_1'(\mathbf{n_s}, \mathbf{0}) & \ldots & \phi_i'(\mathbf{n_s}, \mathbf{0}) & \ldots & \phi_{n_s}'(\mathbf{n_s}, \mathbf{0}) \end{bmatrix}$$

*is invertible, where*

$$\phi_i'(\mathbf{k}, \mathbf{0}) := \frac{\partial \log(p_{\mathbf{z}|\mathbf{u}}(z_i|\mathbf{u_k}))}{\partial z_i} - \frac{\partial \log(p_{\mathbf{z}|\mathbf{u}}(z_i|\mathbf{u_0}))}{\partial z_i}$$

*is the difference of first-order derivative of log density of $z_i$ between domain $\mathbf{u_k}$ and domain $\mathbf{u_0}$ respectively. Then, by learning the estimation $\hat{g}, \hat{\mathbf{z}}_c, \hat{\mathbf{z}}_s$ to achieve equation 3, $\mathbf{z}_s$ is subspace identifiable.*

The proof can be found in Appendix A1.1. Basically, Theorem 1 proposes a weaker form of identifiability with relaxed conditions. With $n_s + 1$ different domains, each true changing variable can be expressed as a function of all estimated changing variables. This indicates that the estimated changing variables capture all information for the true changing variables, and thus disentangle changing and invariant variables. It is imperative to emphasize that, within our framework, the subspace identifiability of changing variables can lead to block-wise identifiability (Kong et al., 2022; von Kügelgen et al., 2021). We provide detailed proof of this in Appendix A1.1. Moreover, it is worth noting that if there is only one changing variable, such subspace identifiability can lead

to component-wise level. When integrated with the continual learning scenario, we uncover the following interesting properties.

**New domains may impair original identifiability of partial changing variables.** Consider a toy case where there are three variables with four domains in total as shown in the top case of Figure 2. The first variable $z_1$ changes in domain $\mathbf{u}_1$ and both $z_1$ and $z_2$ change in domain $\mathbf{u}_2$. When considering only domains $\mathbf{u}_0, \mathbf{u}_1$, $z_1$ can achieve subspace identifiability according to Theorem 1.

It is imperative to recognize that, due to the absence of variability in the remaining variables within these domains, this subspace identifiability inherently aligns with component-wise identifiability. However, when considering domains $\mathbf{u_0}, \mathbf{u_1}, \mathbf{u_2}$, the component-wise identifiability for $z_1$ can't be guaranteed anymore, and instead, we can only promise subspace identifiability for both $z_1$ and $z_2$. In this case, information from domain $\mathbf{u}_2$ can be viewed as "noise" for $z_1$. Contrasted with the traditional joint learning setting, where the data of all domains are overwhelmed, the continual learning setting offers a unique advantage. It allows for achieving and maintaining original identifiability, effectively insulating it from the potential "noise" introduced by newly arriving domains. In Section 4, we empirically demonstrate that the causal representation of $z_1$ obtained through continual learning exhibits better identifiability compared to that obtained through joint training. In addition, another straightforward property is discussed in the Appendix A2.2.

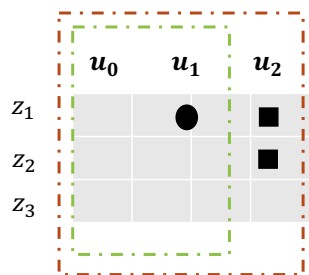

Figure 2: **A toy example with three variables and three domains.** $z_1$ changes in $\mathbf{u}_1, \mathbf{u}_2$, $z_2$ changes in $\mathbf{u}_2$

### 3.3 METHOD

In this section, we leverage the insight of the identifiability theory from previous section to develop our estimation method.

**Generative model.** As shown in Lemma 1 and Theorem 1, we are aiming at estimating causal process $\hat{g}, \hat{\mathbf{z}}_c, \hat{\mathbf{z}}_s$ to reconstruct the distribution of observation. As shown in Figure 3, we construct a Variational Autoencoder (VAE) with its encoder $q_{\hat{g}_\mu^{-1}, \hat{g}_\Sigma^{-1}}(\hat{\mathbf{z}}|\mathbf{x})$ to simulate the mixing process and the decoder $\hat{g}$ to reconstruct a matched distribution $\hat{\mathbf{x}} = \hat{g}(\hat{\mathbf{z}})$. Besides, as introduced in data generation in Equation 1, the changing latent variable is generated as the function of high-level invariance $\hat{\tilde{\mathbf{z}}}_s$ with a specific domain influence $\mathbf{u}$. Assuming the function is invertible, we employ a flow model to obtain the high-level variable $\hat{\tilde{\mathbf{z}}}_s$ by inverting the function, i.e., $\hat{\tilde{\mathbf{z}}}_s = \hat{f}_{\mathbf{u}}^{-1}(\hat{\mathbf{z}}_s)$. To train this model, we apply an ELBO loss as:

$$\mathcal{L}(\hat{g}_\mu^{-1}, \hat{g}_\Sigma^{-1}, \hat{f}_{\mathbf{u}}, \hat{g}) = \mathbb{E}_{\mathbf{x}}\mathbb{E}_{\hat{\mathbf{z}} \sim q_{\hat{g}_\mu^{-1}, \hat{g}_\Sigma^{-1}}} \frac{1}{2}\|x - \hat{x}\|^2 + \alpha KL(q_{\hat{g}_\mu^{-1}, \hat{g}_\Sigma^{-1}}(\hat{\mathbf{z}}_c|\mathbf{x})\|p(\mathbf{z}_c))$$
$$+ \beta KL(q_{\hat{g}_\mu^{-1}, \hat{g}_\Sigma^{-1}, \hat{f}_{\mathbf{u}}}(\hat{\tilde{\mathbf{z}}}_s|\mathbf{x})\|p(\tilde{\mathbf{z}}_s)), \tag{5}$$

where $\alpha$ and $\beta$ are hyperparameters controlling the factor as introduced in (Higgins et al., 2017). To make the equation 5 tractable, we choose the prior distributions $p(\tilde{\mathbf{z}}_s)$ and $p(\mathbf{z}_c)$ as standard Gaussian $\mathcal{N}(\mathbf{0}, \mathbf{I})$.

**Continual casual representation learning.** The subspace identifiability theory in Section 3.2.2 implies that the ground-truth solution lies on a manifold that can be further constrained with more side information, up to the solution with component-wise identifiability. Consequently, it is intuitive to expect that when we observe domains sequentially, the solution space should progressively narrow down in a reasonable manner.

It motivates us to first learn a local solution with existing domains and further improve it to align with the new arriving domain without destroying the original capacity. Specifically, to realize causal representation learning, we employ two objectives: (1) the reconstruction of observations within the current domain, and (2) the preservation of reconstruction capabilities for preceding domains. In terms of implementation, this implies that the movement of network parameters learning a new domain should not result in an increased loss for the previous domains.

To achieve this goal, we found the classical technique GEM (Lopez-Paz & Ranzato, 2017) enables constraining the gradient update of network training to memorize knowledge from previous domains.

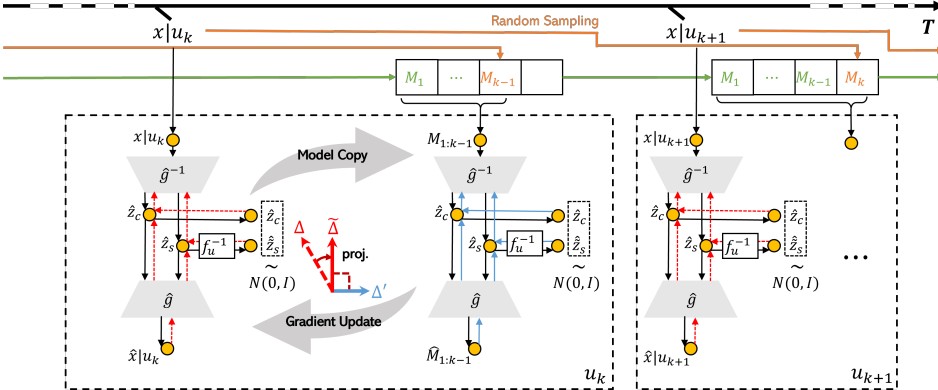

Figure 3: **Overall framework**. For the data from new domain $\mathbf{x}|\mathbf{u}_i$, we calculate the gradients $\Delta$ and $\Delta'$ of our model with both current data and previous memory. Then, we project the gradient $\Delta$ to $\tilde{\Delta}$ using Equation 7 when the angle between $\Delta$ and $\Delta'$ is larger than 90 degrees. Finally, we randomly sample a part of the data in the current domain and add them to the memory bank.

The basic intuition of the algorithm can be illustrated with the following toy example: suppose data from those two domains are denoted as $\{\mathbf{x}|\mathbf{u_1}, \mathbf{x}|\mathbf{u_2}\}$ and the parameter of the network $\boldsymbol{\theta}$ and the loss calculated on data from $k$th domain is denoted as $l(\boldsymbol{\theta}, \mathbf{x}|\mathbf{u_k})$. At the moment of finishing the learning of the first domain, if we don't make any constraints, the model should start the training using data from the second domain with the direction $\frac{\partial l(\boldsymbol{\theta}, \mathbf{x}|\mathbf{u_2})}{\partial \boldsymbol{\theta}}$.

At this moment, if the direction $\frac{\partial l(\boldsymbol{\theta}, \mathbf{x}|\mathbf{u_2})}{\partial \boldsymbol{\theta}}$ happens to have the property that $\langle \frac{\partial l(\boldsymbol{\theta}, \mathbf{x}|\mathbf{u_2})}{\partial \boldsymbol{\theta}}, \frac{\partial l(\boldsymbol{\theta}, \mathbf{x}|\mathbf{u_1})}{\partial \boldsymbol{\theta}} \rangle >$ 0, the current direction will contribute to both domains and we remain the direction. Once the $\langle \frac{\partial l(\boldsymbol{\theta}, \mathbf{x}|\mathbf{u_2})}{\partial \boldsymbol{\theta}}, \frac{\partial l(\boldsymbol{\theta}, \mathbf{x}|\mathbf{u_1})}{\partial \boldsymbol{\theta}} \rangle < 0$ happens, we project the $\frac{\partial l(\boldsymbol{\theta}, \mathbf{x}|\mathbf{u_2})}{\partial \boldsymbol{\theta}}$ to the direction where $\langle \frac{\partial l(\boldsymbol{\theta}, \mathbf{x}|\mathbf{u_2})}{\partial \boldsymbol{\theta}}, \frac{\partial l(\boldsymbol{\theta}, \mathbf{x}|\mathbf{u_1})}{\partial \boldsymbol{\theta}} \rangle = 0$, the orthogonal direction to $\frac{\partial l(\boldsymbol{\theta}, \mathbf{x}|\mathbf{u_1})}{\partial \boldsymbol{\theta}}$ where no loss increment for previous domains. However, there are infinite possible directions satisfying the orthogonal direction requirement. e.g., we can always use the vector containing all zeros. To make the projected gradient as close as the original gradient, we solve for the projected gradient $\frac{\partial l(\boldsymbol{\theta}, \mathbf{x}|\mathbf{u_2})}{\partial \boldsymbol{\theta}}'$ that minimizes the objective function

$$\left\| \frac{\partial l(\boldsymbol{\theta}, \mathbf{x}|\mathbf{u_2})}{\partial \boldsymbol{\theta}} - \frac{\partial l(\boldsymbol{\theta}, \mathbf{x}|\mathbf{u_2})}{\partial \boldsymbol{\theta}}' \right\|^2 \quad \text{s.t.} \quad \frac{\partial l(\boldsymbol{\theta}, \mathbf{x}|\mathbf{u_2})}{\partial \boldsymbol{\theta}}^T \frac{\partial l(\boldsymbol{\theta}, \mathbf{x}|\mathbf{u_1})}{\partial \boldsymbol{\theta}}' \geq 0. \tag{6}$$

Extend into the general case for multiple domains, we consider the following quadratic programming problem w.r.t. vector $\mathbf{v}'$:

$$\min_{\mathbf{v}'} \|\mathbf{v} - \mathbf{v}'\|^2 \quad \text{s.t.} \quad \mathbf{B}\mathbf{v}' \geq 0, \tag{7}$$

where $\mathbf{v}$ denotes the original gradient, $\mathbf{v}'$ denotes the projected gradient, $\mathbf{B}$ is the matrix storing all gradients of past domains. Note that practically, we only store a small portion of data for each domain, thus $\mathbf{B}_i$ is the row of $\mathbf{B}$ storing the memory gradient $\frac{\partial l(\boldsymbol{\theta}, \mathbf{M}|\mathbf{u_i})}{\partial \boldsymbol{\theta}}$, where $\mathbf{M}|\mathbf{u_i} \in \mathbf{x}|\mathbf{u_i}$. We provide the complete procedure in Appendix A3.

## 4 EXPERIMENTS

In this section, we present the implementing details of our method, the experimental results, and the corresponding analysis.

### 4.1 EXPERIMENT SETUP

**Data.** We follow the standard practice employed in previous work (Hyvarinen et al., 2018; Kong et al., 2022) and compare our method to the baselines on synthetic data. We generate the latent variables $\mathbf{z}_s$ for both non-stationary Gaussian and mixed Gaussian with domain-influenced variance and mean, while $\mathbf{z}_c$ for standard Gaussian and mixed Gaussian with constant mean and variance. The

mixing function is estimated by a 2-layer Multi-Layer Perception(MLP) with Leaky-Relu activation. More details can be found in Appendix A5.

**Evaluation metrics.** We use Mean Correlation Coefficient (MCC) to measure the identifiability of the changing variable $\mathbf{z}_s$. However, as the identifiability result can only guarantee component-wise identifiability, it may not be fair to directly use MCC between $\hat{\mathbf{z}}_s$ and $\mathbf{z}_s$ (e.g. if $\hat{\mathbf{z}}_s = \mathbf{z}_s^3$, we will get a distorted MCC value). We thus separate the test data into the training part and test part, and further train separate MLP to learn a simple regression for each $\hat{\mathbf{z}}_s$ to $\mathbf{z}_s$ to remove its nonlinearity on the training part and compute the final MCC on the test part. We repeat our experiments over 5 or 3 random seeds for different settings.

## 4.2 EXPERIMENTAL RESULTS

**Comparison to baseline and joint training.** We evaluate the efficacy of our proposed approach by comparing it against the same model trained on sequentially arriving domains and multiple domains simultaneously, referred to as the baseline and theoretical upper bound by the continual learning community. We employ identical network architectures for all three models and examine four distinct datasets, with respective parameters of $\mathbf{z}_s$ being Gaussian and mixed Gaussian with $n_s = 4, n = 8$, as well as $n_s = 2, n = 4$. Increasing numbers of domains are assessed for each dataset. Figure 4 shows our method reaches comparable performance with joint training. Further visualization can be found in Appendix A4.

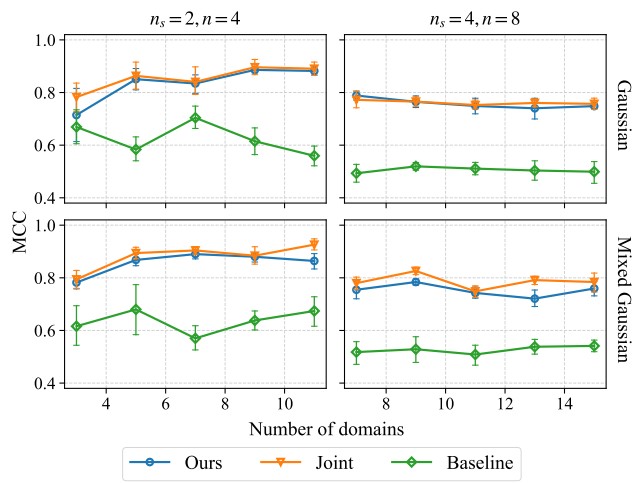

Figure 4: Comparison of MCC for all four datasets with the number of domains from $2n_s - 1$ to $2n_s + 7$. In this instance, the number of training and the number of testing domains are equated, which differs from the investigation for increasing domains.

**Increasing domains.** For dataset $n_s = 4, n = 8$ of Gaussian, we save every trained model after each domain and evaluate their MCC. Specifically, we evaluated the models on the original test dataset, which encompasses data from all 15 domains. As shown in part (a) of Figure 5, remarkably, increasing domains lead to greater identifiability results, which align with our expectations that sequential learning uncovers the true underlying causal relationships as more information is revealed. Specifically, we observe that the MCC reaches a performance plateau at 9 domains and the

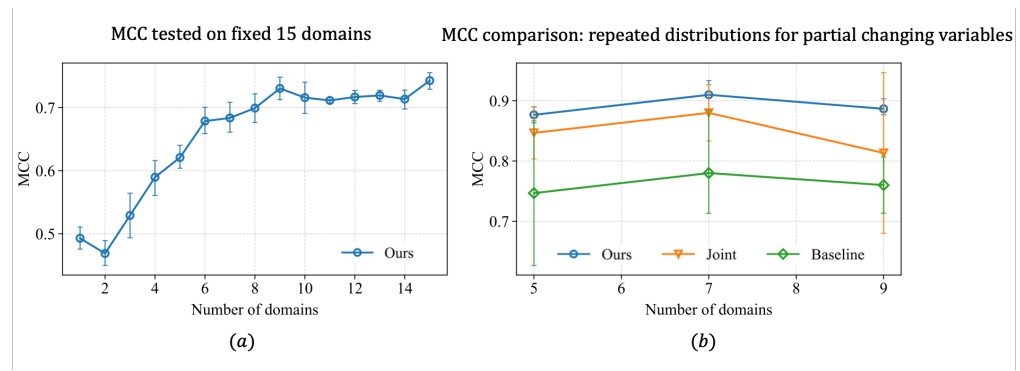

Figure 5: (a) MCC for increasing domains with models tested on all 15 domains after training of each domain. (b) MCC for models trained with repeated distributions for partial changing variables.

extra domains(from 9 to 15) don't provide further improvement. This appears to be consistent with the identifiability theory that $2n_s + 1 = 9$ domains are needed for identifiability.

**Repeated distributions for partial changing variables.** For dataset $n_s = 2, n = 4$ of Gaussian, we test the case that $z_{s,1}$ have changing distributions over all domains while $z_{s,2}$ only holds three different distributions across domains. As shown in part (b) of Figure 5, our method outperforms both joint train and baseline. It may be because our method has the ability to maintain the performance learned from previous domains and prevents potential impairment from new arriving domains with repeated distributions. For this instance, our method exhibits more robust performance than joint training against negative effects from $8, 9$th domains.

**Discussion: is joint training always better than learning sequentially?  Not necessarily.** As discussed in Section 3.2.2, the new domain may impair the identifiability of partial variables. While joint training always shuffles the data and doesn't care about the order information, learning sequentially to some extent mitigates the impairment of identifiability.

Specifically, we conducted an experiment in which both $z_1$ and $z_2$ are Gaussian variables. The variance and mean of $z_1$ change in the second domain, while the other variable changes in the third domain. We then compare our method with joint training only for latent variable $z_1$. We repeat our experiments with 3 random seeds and the experiment shows that the MCC of our method for $z_1$ reaches up to 0.785 while joint training retains at 0.68 as shown in Figure 6. In terms of visual contrast, the scatter plot obtained using our method on the left of Figure 6 exhibits a significantly stronger linear correlation compared to joint training. Further discussions and propositions are provided in Appendix A2.1

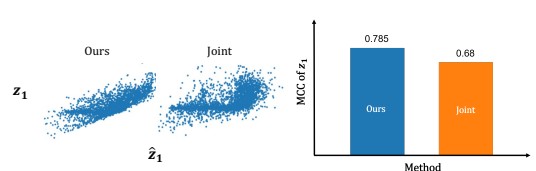

Figure 6: Comparison of identifiability for $z_1$ using Joint training and our method qualitatively and quantitatively.

**Ablation study on prior knowledge of changing variables.** A major limitation of our approach is the requirement for prior knowledge of the number of changing variables. Developing a method to automatically determine the number of changing variables is nontrivial in the continual learning scenario. Therefore, we turn to conducting an ablation study to investigate the sensitivity of this prior knowledge. We implemented the ablation study on mixed Gaussian

|  | T=3 | T=5 | T=7 | T=9 |
|---|---|---|---|---|
| $\hat{n}_s = 2$ | 0.782 | **0.868** | **0.890** | **0.880** |
| $\hat{n}_s = 3$ | 0.781 | 0.835 | 0.836 | 0.834 |
| $\hat{n}_s = 4$ | **0.830** | 0.861 | 0.838 | 0.868 |

Table 1: mean MCC comparison for different preset value of the number of changing variables $\hat{n}_s$

case with $n = 4, n_s = 2$ five times with different seeds. The results as shown in Table 1 indicate that our method exhibits relative stability, with a discernible performance decline observed when there is a mismatch between the actual and estimated numbers.

## 5 CONCLUSION

In this paper, we present a novel approach for learning causal representation in continually arriving domains. Through theoretical analysis, we have examined the relationship between model identification and the number of observed domains. Our findings indicate that as additional domains are incorporated, the identifiability of changing variables escalates, with subspace identification achievable with $n_s + 1$ domains and component-wise identification requiring $2n_s + 1$ domains or more. Besides, we briefly show that a carefully chosen order of learning leads to meaningful disentanglement after each domain is learned, and the introduction of new domains does not necessarily contribute to all variables. To realize CCRL, we employed GEM to preserve prior knowledge and refine it using information derived from incoming domains, resembling human learning mechanisms. Empirical evaluations have demonstrated that our approach achieves performance on par with nonlinear ICA techniques trained jointly across multiple offline domains, exhibiting greater identifiability with increasing domains observed.

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

*Appendix for*

## "Continual Nonlinear ICA-based Representation Learning"

Appendix organization:

---

---

## A1   PROOF AND DISCUSSION

We divide our proof into the following parts. First, we start from the matched distribution of the estimated observation and the real observation, then we show the the true latent variables can be expressed as invertible transformations of the estimated variables. We then use derivatives to construct component-wise relations between the estimated variables with the true latents. We finally show, with enough domains, we can construct the matrix whose invertibility will force the changing variables subspace identifiable with $n_s + 1$ domains and component-wise identifiable with $2n_s + 1$ domains.

We start from the matched distribution as introduced in Equation 3: for $\mathbf{u}' \in \mathcal{U}$

$$p_{\mathbf{x}|\mathbf{u}} = p_{\hat{\mathbf{x}}|\mathbf{u}}. \tag{8}$$

will imply

$$p_{g(\mathbf{z})|\mathbf{u}} = p_{\hat{g}(\hat{\mathbf{z}})|\mathbf{u}}. \tag{9}$$

according to the function of transformation, we can get

$$p_{g^{-1}\circ g(\mathbf{z})|\mathbf{u}}|J_g^{-1}| = p_{g^{-1}\circ \hat{g}(\hat{\mathbf{z}})|\mathbf{u}}|J_g^{-1}|. \tag{10}$$

Let $h := g^{-1} \circ \hat{g}$ to express the transformation from estimated latent variables to real latent variables, i.e., $\mathbf{z} = h(\hat{\mathbf{z}})$. As long as both $\hat{g}$ and $g$ are invertible, the transformation $h$ should also be invertible. We can then get the following

$$p_{\mathbf{z}|\mathbf{u}} = p_{h(\hat{\mathbf{z}})|\mathbf{u}}. \tag{11}$$

according to the conditional independence(assumption 3) and nonzero densities(assumption 1) in Lemma 1, the log density of each marginal distribution can be expressed as

$$\log p_{\mathbf{z}|\mathbf{u}}(\mathbf{z}) = \sum_{i=1}^{n} \log p_{z_i|\mathbf{u}}(z_i);$$
$$\log p_{\hat{\mathbf{z}}|\mathbf{u}}(\mathbf{z}) = \sum_{i=1}^{n} \log p_{\hat{z}_i|\mathbf{u}}(\hat{z}_i).$$
(12)

Thus, from Equation 11 and according to the function of transformation

$$p_{\mathbf{z}|\mathbf{u}} = p_{(\hat{\mathbf{z}})|\mathbf{u}}|J_{h^{-1}}|.$$
(13)

Take log density on both sides,

$$\sum_{i=1}^{n} \log p_{z_i|\mathbf{u}}(z_i) = \sum_{i=1}^{n} \log p_{\hat{z}_i|\mathbf{u}}(\hat{z}_i) + \log|J_{h^{-1}}|.$$
(14)

Simplify the notation as $q_i(z_i, \mathbf{u}) = \log p_{z_i|\mathbf{u}}(z_i), \hat{q}_i(\hat{z}_i, \mathbf{u}) = \log p_{\hat{z}_i|\mathbf{u}}(\hat{z}_i)$, the above equation is

$$\sum_{i=1}^{n} q_i(z_i, \mathbf{u}) = \sum_{i=1}^{n} \hat{q}_i(\hat{z}_i, \mathbf{u}) + \log|J_{h^{-1}}|.$$
(15)

From Equation 15, we can see

$$\sum_{i=1}^{n} q_i(z_i, \mathbf{u}) + \log|J_h| = \sum_{i=1}^{n} \hat{q}_i(\hat{z}_i, \mathbf{u})$$
(16)

Until now, we have constructed the relationship between all true latent variables and all estimated variables. In the following sections, we will show how to use the technique of derivatives to establish component-wise relationships between them and how to utilize multi-domain information to eliminate the intractable Jacobian term.

In Section A1.1, we show the proof of Theorem 1 and the Proposition 1 inspired by it. In Section A1.2, we show the proof of Lemma 1. In Section A1.3, we discuss the case where there are repeated distributions across different domains for partial changing variables and show if there are two or more changing variables, at least three non-repetitive distributions are required for each variable.

### A1.1 SUBSPACE IDENTIFIABILITY WITH $n_s + 1$ DOMAINS

Take the derivative of Equation 16 with estimated invariant variable $\hat{z}_j$ where $j \in \{n_{s+1}, \ldots, n\}$. We can get

$$\sum_{i=1}^{n} \frac{\partial q_i(z_i, \mathbf{u})}{\partial z_i} \frac{\partial z_i}{\partial \hat{z}_j} + \frac{\partial \log|J_h|}{\partial \hat{z}_j} = \frac{\partial \hat{q}_j(\hat{z}_j, \mathbf{u})}{\partial \hat{z}_j}$$
(17)

The equation allows us to construct the component-wise relation between true latent variable $\mathbf{z}$ with estimated invariant variables $\hat{\mathbf{z}}_c$ as expressed using $\frac{\partial z_i}{\partial \hat{z}_j}$. However, the Jacobian term $\frac{\partial \log|J_h|}{\partial \hat{z}_j}$ is intractable as we have no knowledge about $h$(once we have, everything is solved). If we have multiple domains $\mathbf{u} = \mathbf{u_0}, \ldots, \mathbf{u_{n_s}}$, we have $n_s + 1$ equations like equation above. We can remove the intractable Jacobian by taking the difference for every equation $\mathbf{u} = \mathbf{u_1}, \ldots, \mathbf{u_{n_s}}$ with the equation where $\mathbf{u} = \mathbf{u_0}$:

$$\sum_{i=1}^{n} \left( \frac{\partial q_i(z_i, \mathbf{u_q})}{\partial z_i} - \frac{\partial q_i(z_i, \mathbf{u_0})}{\partial z_i} \right) \frac{\partial z_i}{\partial \hat{z}_j} = \frac{\partial \hat{q}_j(\hat{z}_j, \mathbf{u_q})}{\partial \hat{z}_j} - \frac{\partial \hat{q}_j(\hat{z}_j, \mathbf{u_0})}{\partial \hat{z}_j}$$
(18)

As long as the $j \in \{n_{s+1}, \ldots, n\}$, the distribution of estimated variable $\hat{z}_j$ doesn't change across all domains. The right-hand side of the equation above will be zero. Thus,

$$\sum_{i=1}^{n} \left( \frac{\partial q_i(z_i, \mathbf{u_k})}{\partial z_i} - \frac{\partial q_i(z_i, \mathbf{u_0})}{\partial z_i} \right) \frac{\partial z_i}{\partial \hat{z}_j} = 0$$
(19)

Similarly, $q_i(z_i, \mathbf{u})$ remains the same for $i \in \{n_{s+1}, \ldots, n\}$

$$\sum_{i=1}^{n_s} \left( \frac{\partial q_i(z_i, \mathbf{u_k})}{\partial z_i} - \frac{\partial q_i(z_i, \mathbf{u_0})}{\partial z_i} \right) \frac{\partial z_i}{\partial \hat{z}_j} = 0 \tag{20}$$

Thus, we can have the linear system:

$$\begin{bmatrix} \frac{\partial q_1(z_1, \mathbf{u_1})}{\partial z_1} - \frac{\partial q_1(z_1, \mathbf{u_0})}{\partial z_1} & \cdots & \frac{\partial q_{n_s}(z_{n_s}, \mathbf{u_1})}{\partial z_{n_s}} - \frac{\partial q_{n_s}(z_{n_s}, \mathbf{u_0})}{\partial z_{n_s}} \\ \vdots & \cdots & \vdots \\ \frac{\partial q_1(z_1, \mathbf{u_{n_s}})}{\partial z_1} - \frac{\partial q_1(z_1, \mathbf{u_0})}{\partial z_1} & \cdots & \frac{\partial q_{n_s}(z_{n_s}, \mathbf{u_{n_s}})}{\partial z_{n_s}} - \frac{\partial q_{n_s}(z_{n_s}, \mathbf{u_0})}{\partial z_{n_s}} \end{bmatrix} \begin{bmatrix} \frac{\partial z_1}{\partial \hat{z}_j} \\ \vdots \\ \frac{\partial z_{n_s}}{\partial \hat{z}_j} \end{bmatrix} = 0 \tag{21}$$

If the matrix above is invertible, its null space will only contain all zeros. Thus, $\frac{\partial z_i}{\partial \hat{z}_j} = 0$ for any $i \in \{1, \ldots, n_s\}, j \in \{n_{s+1}, \ldots, n\}$. That is, $\frac{\partial \mathbf{z}_s}{\partial \hat{\mathbf{z}}_c} = 0$. Simplify the notation and define

$$\phi_i'(\mathbf{k}) := \frac{\partial \log(p_{\mathbf{z}|\mathbf{u}}(z_i|\mathbf{u_k}))}{\partial z_i} - \frac{\partial \log(p_{\mathbf{z}|\mathbf{u}}(z_i|\mathbf{u_0}))}{\partial z_i}$$

If the matrix

$$\begin{bmatrix} \phi_1'(\mathbf{1}) & \cdots & \phi_i'(\mathbf{1}) & \cdots & \phi_{n_s}'(\mathbf{1}) \\ \vdots & \ddots & \vdots & \vdots & \vdots \\ \phi_1'(\mathbf{k}) & \cdots & \phi_i'(\mathbf{k}) & \cdots & \phi_{n_s}'(\mathbf{k}) \\ \vdots & \vdots & \vdots & \ddots & \vdots \\ \phi_1'(\mathbf{n_s}) & \cdots & \phi_i'(\mathbf{n_s}) & \cdots & \phi_{n_s}'(\mathbf{n_s}) \end{bmatrix} \tag{23}$$

is invertible, we can get $\frac{\partial \mathbf{z}_s}{\partial \hat{\mathbf{z}}_c} = 0$.

We further look back into the Jacobian matrix which captures the relation true latent variables $\mathbf{z}$ with the estimated variables $\mathbf{z}$:

$$J_h = \begin{bmatrix} \frac{\partial \mathbf{z}_c}{\partial \hat{\mathbf{z}}_c} & \frac{\partial \mathbf{z}_c}{\partial \hat{\mathbf{z}}_s} \\ \frac{\partial \mathbf{z}_s}{\partial \hat{\mathbf{z}}_c} & \frac{\partial \mathbf{z}_s}{\partial \hat{\mathbf{z}}_s} \end{bmatrix}. \tag{24}$$

As long as the transformation $h$ is invertible, the Jacobian matrix $J_h$ should be full rank. Thus, The $\frac{\partial \mathbf{z}_s}{\partial \hat{\mathbf{z}}_c}$ means that the bottom row of Jacobian above can only contain non-zero in $\frac{\partial \mathbf{z}_s}{\partial \hat{\mathbf{z}}_s}$. That is, for each true changing variable $z_{s,i}$, it can be written as the function $h_i$ of the estimated changing variables $\hat{\mathbf{z}}_s$ such that $z_{s,i} = h_i(\hat{\mathbf{z}}_s)$, which accomplishes the proof.

**Proposition 1** *If Theorem 1 holds, for the estimated changing variables $\hat{\mathbf{z}}_s$ and true changing variables $\mathbf{z}_s$, there exist an invertible function $h_s : \mathcal{R}^{n_s} \to \mathcal{R}^{n_s}$ such that $\mathbf{z}_s = \hat{\mathbf{z}}_s$ (**block-wise identifiability**).*

*Proof* We follow the result that $\frac{\partial \mathbf{z}_s}{\partial \hat{\mathbf{z}}_c} = 0$ and recall the large Jocabian matrix $J_h$ is invertible, according to the property of invertible block matrix that

$$\det \begin{bmatrix} A & B \\ 0 & D \end{bmatrix} = \det(A) \det(D). \tag{25}$$

Thus, we can derive the determinant of $J_h$ is

$$\det(J_h) = \det(\frac{\partial \mathbf{z}_c}{\partial \hat{\mathbf{z}}_c}) \det(\frac{\partial \mathbf{z}_s}{\partial \hat{\mathbf{z}}_s}). \tag{26}$$

As long as $\det(J_h) \neq 0$ ($J_h$ is full rank), neither $\det(\frac{\partial \mathbf{z}_c}{\partial \hat{\mathbf{z}}_c})$ nor $\det(\frac{\partial \mathbf{z}_s}{\partial \hat{\mathbf{z}}_s})$ should equal to 0. Thus, the transformation from $\mathbf{z}_s$ to $\hat{\mathbf{z}}_s$ should be invertible, which accomplishes the proof.

**Remark 2** *We know that $\frac{\partial \mathbf{z}_s}{\partial \hat{\mathbf{z}}_c} = 0$, which is kind of trivial intuitively as $z_s$ is changing while $\hat{z}_c$ remains the same distribution. However, as the Jacobian matrix $J_h$ is invertible, we can utilize its property that the inverse of a block matrix is*

$$\begin{bmatrix} A & B \\ C & D \end{bmatrix}^{-1} = \begin{bmatrix} (A - BD^{-1}C)^{-1} & -(A - BD^{-1}C)^{-1}BD^{-1} \\ -D^{-1}C(A - BD^{-1}C)^{-1} & D^{-1} + D^{-1}C(A - BD^{-1}C)^{-1}BD^{-1} \end{bmatrix} \quad (27)$$

*Thus, for the inverse of the Jacobian matrix above*

$$J_h^{-1} = \begin{bmatrix} \frac{\partial \hat{\mathbf{z}}_c}{\partial \mathbf{z}_c} & \frac{\partial \hat{\mathbf{z}}_c}{\partial \mathbf{z}_s} \\ \frac{\partial \hat{\mathbf{z}}_s}{\partial \mathbf{z}_c} & \frac{\partial \hat{\mathbf{z}}_s}{\partial \mathbf{z}_s} \end{bmatrix} \quad (28)$$

*The bottom left term $\frac{\partial \hat{\mathbf{z}}_s}{\partial \mathbf{z}_c}$ must be zero. This provides more valuable insight, stating that the estimated changing variables cannot be expressed as the function of true invariant variables.*

### A1.2 COMPONENT-WISE IDENTIFIABILITY FOR $2n_s + 1$ DOMAINS

Differentiating both sides of Equation 16 with respect to $\hat{z}_j$, $j \in \{1, \ldots, n\}$, we can get

$$\frac{\partial \hat{q}_j(\hat{z}_j, \mathbf{u})}{\partial \hat{z}_j} = \sum_{i=1}^{n} \frac{\partial q_i(z_i, \mathbf{u})}{\partial z_i} \frac{\partial z_i}{\partial \hat{z}_j} + \frac{\partial \log |J_h|}{\partial \hat{z}_j}. \quad (29)$$

Further differentiate with respect to $\hat{z}_q$, $q \in \{1, \ldots, n\}, q \neq j$, according to the chain rule,

$$0 = \sum_{i=1}^{n} \frac{\partial^2 q_i(z_i, \mathbf{u})}{\partial z_i^2} \frac{\partial z_i}{\partial \hat{z}_j} \frac{\partial z_i}{\partial \hat{z}_q} + \frac{\partial q_i(z_i, \mathbf{u})}{\partial z_i} \frac{\partial^2 z_i}{\partial \hat{z}_j \partial \hat{z}_q} + \frac{\partial^2 \log |J_h|}{\partial \hat{z}_j \partial \hat{z}_q}. \quad (30)$$

This equation allows us to have the component-wise relation between $\hat{\mathbf{z}}$ with $\mathbf{z}$. Following the same ideas, and introducing multiple domains come into play to remove the Jacobian term. Using assumption 4 in Lemma1, for $\mathbf{u} = \mathbf{u_0}, \ldots, \mathbf{u_{2n_s}}$, we have $2n_s + 1$ equations like Equation 30. Therefore, we can remove the effect of the Jacobian term by taking the difference for every equation $\mathbf{u} = \mathbf{u_1}, \ldots, \mathbf{u_{2n_s}}$ with the equation where $\mathbf{u} = \mathbf{u_0}$:

$$\sum_{i=1}^{n} \left(\frac{\partial^2 q_i(z_i, \mathbf{u_k})}{\partial z_i^2} - \frac{\partial^2 q_i(z_i, \mathbf{u_0})}{\partial z_i^2}\right) \frac{\partial z_i}{\partial \hat{z}_j} \frac{\partial z_i}{\partial \hat{z}_q} + \left(\frac{\partial q_i(z_i, \mathbf{u_k})}{\partial z_i} - \frac{\partial q_i(z_i, \mathbf{u_0})}{\partial z_i}\right) \frac{\partial^2 z_i}{\partial \hat{z}_j \partial \hat{z}_q} = 0. \quad (31)$$

For invariant variables $\mathbf{z}_c$, their log density doesn't change across different domains. Thus, we can get rid of invariant parts of the equation above and have

$$\sum_{i=1}^{n_s} \left(\frac{\partial^2 q_i(z_i, \mathbf{u_k})}{\partial z_i^2} - \frac{\partial^2 q_i(z_i, \mathbf{u_0})}{\partial z_i^2}\right) \frac{\partial z_i}{\partial \hat{z}_j} \frac{\partial z_i}{\partial \hat{z}_q} + \left(\frac{\partial q_i(z_i, \mathbf{u_k})}{\partial z_i} - \frac{\partial q_i(z_i, \mathbf{u_0})}{\partial z_i}\right) \frac{\partial^2 z_i}{\partial \hat{z}_j \partial \hat{z}_q} = 0. \quad (32)$$

Simplify the notation as $\phi_i''(\mathbf{k}) := \frac{\partial^2 q_i(z_i, \mathbf{u_k})}{\partial z_i^2} - \frac{\partial^2 q_i(z_i, \mathbf{u_0})}{\partial z_i^2}$ , $\phi_i'(\mathbf{k}) := \frac{\partial q_i(z_i, \mathbf{u_k})}{\partial z_i} - \frac{\partial q_i(z_i, \mathbf{u_0})}{\partial z_i}$ and rewrite those equations above as a linear system, we have

$$\begin{bmatrix} \phi_1''(\mathbf{1}) & \cdots & \phi_i''(\mathbf{1}) & \cdots & \phi_{n_s}''(\mathbf{1}) & \phi_1'(\mathbf{1}) & \cdots & \phi_i'(\mathbf{1}) & \cdots & \phi_{n_s}'(\mathbf{1}) \\ \vdots & \ddots & \vdots & \vdots & \vdots & \vdots & \vdots & \ddots & \vdots & \vdots \\ \phi_1''(\mathbf{k}) & \cdots & \phi_i''(\mathbf{k}) & \cdots & \phi_{n_s}''(\mathbf{k}) & \phi_1'(\mathbf{k}) & \cdots & \phi_i'(\mathbf{k}) & \cdots & \phi_{n_s}'(\mathbf{k}) \\ \vdots & \ddots & \vdots & \vdots & \vdots & \vdots & \vdots & \ddots & \vdots & \vdots \\ \phi_1''(\mathbf{2n_s}) & \cdots & \phi_i''(\mathbf{2n_s}) & \cdots & \phi_{n_s}''(\mathbf{2n_s}) & \phi_1'(\mathbf{2n_s}) & \cdots & \phi_i'(\mathbf{2n_s}) & \cdots & \phi_{n_s}'(\mathbf{2n_s}) \end{bmatrix} \begin{bmatrix} \frac{\partial z_1}{\partial \hat{z}_j} \frac{\partial z_1}{\partial \hat{z}_q} \\ \vdots \\ \frac{\partial z_{n_s}}{\partial \hat{z}_j} \frac{\partial z_{n_s}}{\partial \hat{z}_q} \\ \frac{\partial^2 z_1}{\partial \hat{z}_j \hat{z}_q} \\ \vdots \\ \frac{\partial^2 z_{n_s}}{\partial \hat{z}_j \hat{z}_q} \end{bmatrix} = \mathbf{0}.$$

$$(33)$$

Thus, if the above matrix is invertible according to assumption 4 in Theorem 1, we will leave its null space all zero. i.e., $\frac{\partial z_i^2}{\partial \hat{z}_j \hat{z}_q} = 0$ and $\frac{\partial z_i}{\partial \hat{z}_j} \frac{\partial z_i}{\partial \hat{z}_q} = 0$ for all $i \in \{1, \ldots, n_s\}, j, q \in \{1, \ldots, n\}, j \neq q$. We further use the property that the $h$ is invertible, which means for the Jacobian matrix of transformation $h$:

$$J_h = \begin{bmatrix} \frac{\partial \mathbf{z}_c}{\partial \hat{\mathbf{z}}_c} & \frac{\partial \mathbf{z}_c}{\partial \hat{\mathbf{z}}_s} \\ \frac{\partial \mathbf{z}_s}{\partial \hat{\mathbf{z}}_c} & \frac{\partial \mathbf{z}_\mathbf{s}}{\partial \hat{\mathbf{z}}_s} \end{bmatrix}. \tag{34}$$

the $[\frac{\partial \mathbf{z}_s}{\partial \hat{\mathbf{z}}_c}, \frac{\partial \mathbf{z}_\mathbf{s}}{\partial \hat{\mathbf{z}}_s}]$ contains only one non zero value in each row. As proven in Appendix A1.1, we can get $\frac{\partial \mathbf{z}_s}{\partial \hat{\mathbf{z}}_c} = 0$ with number of domains larger or equal to $n_s + 1$. Thus, $\frac{\partial \mathbf{z}_\mathbf{s}}{\partial \hat{\mathbf{z}}_s}$ is an invertible full rank-matrix with only one nonzero value in each row. The changing variable $\mathbf{z}_s$ is component-wise identifiable.

### A1.3 DISCUSSION OF COMPONENT-WISE IDENTIFIABILITY OF REPEATED DISTRIBUTION FOR PARTIAL CHANGING VARIABLES

In this section, we start with an example to discuss the possible scenarios where there are repeated distributions for partially changing variables among different domains. Based on this example, we proceed to provide an intuitive proof of Remark 1.

Let's follow the proof of component-wise identifiability of changing variables. We directly look into the equation

$$0 = \sum_{i=1}^{n} \frac{\partial^2 q_i(z_i, \mathbf{u})}{\partial z_i^2} \frac{\partial z_i}{\partial \hat{z}_j} \frac{\partial z_i}{\partial \hat{z}_q} + \frac{\partial q_i(z_i, \mathbf{u})}{\partial z_i} \frac{\partial^2 z_i}{\partial \hat{z}_j \partial \hat{z}_q} + \frac{\partial^2 \log |J_h|}{\partial \hat{z}_j \partial \hat{z}_q}. \tag{35}$$

Our goal is to produce the matrix containing $\frac{\partial^2 q_i(z_i, \mathbf{u})}{\partial z_i^2}$ and $\frac{\partial q_i(z_i, \mathbf{u})}{\partial z_i}$ whose null space only contains zero vector. However, we can't ensure every arrived domain will bring enough change. In this case, distributions of the same variable on different domains may be the same. i.e., $q_i(z_i, \mathbf{u_l}) = q_i(z_i, \mathbf{u_k})$ where $l \neq k$. Our discussion will mainly revolve around this situation.

Let's start with the simplest case where there are only two changing variables $z_1$ and $z_2$ and no invariant variables. We know from Theorem1 that we need $2n_s + 1$ domains to reveal their component-wise identifiability. Therefore, for $\mathbf{u} = \mathbf{u_0}, \ldots, \mathbf{u_4}$, we have the following linear system:

$$\begin{bmatrix} \phi_1''(\mathbf{1,0}) & \phi_2''(\mathbf{1,0}) & \phi_1'(\mathbf{1,0}) & \phi_2'(\mathbf{1,0}) \\ \phi_1''(\mathbf{2,0}) & \phi_2''(\mathbf{2,0}) & \phi_1'(\mathbf{2,0}) & \phi_2'(\mathbf{2,0}) \\ \phi_1''(\mathbf{3,0}) & \phi_2''(\mathbf{3,0}) & \phi_1'(\mathbf{3,0}) & \phi_2'(\mathbf{3,0}) \\ \phi_1''(\mathbf{4,0}) & \phi_2''(\mathbf{4,0}) & \phi_1'(\mathbf{4,0}) & \phi_2'(\mathbf{4,0}) \end{bmatrix} \begin{bmatrix} \frac{\partial z_1}{\partial \hat{z}_1} \frac{\partial z_1}{\partial \hat{z}_2} \\ \frac{\partial z_2}{\partial \hat{z}_1} \frac{\partial z_2}{\partial \hat{z}_2} \\ \frac{\partial^2 z_1}{\partial \hat{z}_1 \hat{z}_2} \\ \frac{\partial^2 z_2}{\partial \hat{z}_1 \hat{z}_2} \end{bmatrix} = \mathbf{0}.$$

where $\phi_i''(\mathbf{k,l}) := \frac{\partial^2 q_i(z_i, \mathbf{u_k})}{\partial z_i^2} - \frac{\partial^2 q_i(z_i, \mathbf{u_l})}{\partial z_i^2}$, $\phi_i'(\mathbf{k,l}) := \frac{\partial q_i(z_i, \mathbf{u_k})}{\partial z_i} - \frac{\partial q_i(z_i, \mathbf{u_l})}{\partial z_i}$

Assume $z_1$ varies sufficiently across all domains. i.e., $q_1(z_1, \mathbf{u_j}) \neq q_1(z_1, \mathbf{u_k})$ for all $k, l \in \{1, \ldots, 5\}$, while $z_2$ partially changes across domains, e.g., $q_2(z_2, \mathbf{u_0}) \neq q_2(z_2, \mathbf{u_1}) = q_2(z_2, \mathbf{u_2}) = q_2(z_2, \mathbf{u_3}) = q_2(z_2, \mathbf{u_4})$. Subtract the first row with other rows, we have

$$\begin{bmatrix} \phi_1''(\mathbf{1,0}) & \phi_2''(\mathbf{1,0}) & \phi_1'(\mathbf{1,0}) & \phi_2'(\mathbf{1,0}) \\ \phi_1''(\mathbf{2,1}) & 0 & \phi_1'(\mathbf{2,1}) & 0 \\ \phi_1''(\mathbf{3,1}) & 0 & \phi_1'(\mathbf{3,1}) & 0 \\ \phi_1''(\mathbf{4,1}) & 0 & \phi_1'(\mathbf{4,1}) & 0 \end{bmatrix} \begin{bmatrix} \frac{\partial z_1}{\partial \hat{z}_1} \frac{\partial z_1}{\partial \hat{z}_2} \\ \frac{\partial z_2}{\partial \hat{z}_1} \frac{\partial z_2}{\partial \hat{z}_2} \\ \frac{\partial^2 z_1}{\partial \hat{z}_1 \hat{z}_2} \\ \frac{\partial^2 z_2}{\partial \hat{z}_1 \hat{z}_2} \end{bmatrix} = \mathbf{0}.$$

Apparently, the matrix above is not invertible as the second column and fourth column are dependent. What if we further release the condition by introducing new changing domains? i.e., $q_2(z_2, \mathbf{u_0}) \neq q_2(z_2, \mathbf{u_1}) \neq q_2(z_2, \mathbf{u_2}) = q_2(z_2, \mathbf{u_3}) = q_2(z_2, \mathbf{u_4})$. We will have the following linear system:

$$
\begin{bmatrix}
\phi_1''(\mathbf{1,0}) & \phi_2''(\mathbf{1,0}) & \phi_1'(\mathbf{1,0}) & \phi_2'(\mathbf{1,0}) \\
\phi_1''(\mathbf{2,1}) & \phi_2''(\mathbf{2,1}) & \phi_1'(\mathbf{2,1}) & \phi_2'(\mathbf{2,1}) \\
\phi_1''(\mathbf{3,1}) & 0 & \phi_1'(\mathbf{3,1}) & 0 \\
\phi_1''(\mathbf{4,1}) & 0 & \phi_1'(\mathbf{4,1}) & 0
\end{bmatrix}
\begin{bmatrix}
\frac{\partial z_1}{\partial \hat{z}_1} \frac{\partial z_1}{\partial \hat{z}_2} \\[6pt]
\frac{\partial z_2}{\partial \hat{z}_1} \frac{\partial z_2}{\partial \hat{z}_2} \\[6pt]
\frac{\partial^2 z_1}{\partial \hat{z}_1 \hat{z}_2} \\[6pt]
\frac{\partial^2 z_2}{\partial \hat{z}_1 \hat{z}_2}
\end{bmatrix}
= \mathbf{0}.
$$

From the example above, we can easily prove the Remark 1 from its contra-positive perspective, which is to prove that for $n_s \geq 2$, if $|S_i| \leq 2$, then the Lemma 1 cannot hold. We denote the matrix in Lemma 1.4 as $W$ and use $W_{:,j}$ to denote the $jth$ column of the matrix $W$. For any latent changing variable $z_i$ whose $|S_i| \leq 2$, there are at most two different distributions across all observed domains. Thus, there is at most one nonzero entry in $W_{:,i}$ and $W_{:,2i}$, which will directly lead to the linear dependence between $W_{:,i}$ with $W_{:,2i}$. The Lemma 1 cannot hold.

## A2  DISCUSSION OF THOSE PROPERTIES

### A2.1  POSSIBLE IMPAIRMENT FOR PARTIAL CHANGING VARIABLES WHEN NEW DOMAINS ARE INVOLVED

Before we dive into the details, we need to first clarify one basic concept: **Incremental domains can't affect the overall identifiability of all changing variables theoretically**. As long as both Theorem 1 and Lemma 1 state that the overall identifiability is determined by the distribution of true latent changing variables, the way of learning can't influence it theoretically. However, the identifiability of partial changing variables will be affected as shown in Section 3.2.2 and Experiment 4.2. We demonstrate the influence of the new domain on the identifiability of partial variables through a carefully designed example, as inferred below.

**Proposition 2** *For latent variables $z_1, \ldots, z_n$ with sequentially arriving domains $u_0, \ldots, u_T$ whose generation process follows equation 1 and Lemma1.1,1.2,1.3 hold. Assume for latent variable $z_i, i \in \{1, \ldots, n\}$, the first change happens at $u_{2i-1}$ and the following domains will bring sufficient change, i.e., $p(z_i|u_0) = \cdots = p(z_i|u_{2i-2}) \neq p(z_i|u_{2i-1}) \neq \cdots \neq p(z_i, u_T)$. Then the latent variable $z_i$ reaches component-wise identifiability after observing $u_{2i+2k \leq T}$ where $k = \{0, 1, \ldots |2i + 2k \leq T\}$, and its identifiability degrades to subspace level after the observation of $u_{2i+2k+1}$ where $k = \{0, 1, \ldots |2i + 2k + 1 \leq T\}$.*

*Proof* As long as the distribution of latent variables $z_i$ follows the $p(z_i|u_0) = \cdots = p(z_i|u_{2i-2}) \neq p(z_i|u_{2i-1}) \neq \cdots \neq p(z_i, u_T)$. The latent variable $z_i$ can only be referred to as "changing variable" after observation of $u_{2i-1}$. In other words, before the observation of $u_{2i-1}$, the changing variables only include $\{z_1, ; z_{i-1}\}$. At this moment, we have $\{u_0, \ldots, u_{2i-2}\}$ different domains with sufficient change, and the requirement of Lemma 1.4 is satisfied. The latent variables $\{z_1, \ldots, z_{i-1}\}$ are component-wise identifiable.

When the observation of $u_{2i-1}$ happens, a new changing variable $z_i$ is introduced. The condition in Lemma 1.4 doesn't hold anymore as there are only $2n_s$ domains while $2n_s + 1$ are required. However, the conditions in Theorem 1 are still fulfilled, thus the identifiability of $\{z_1, \ldots, z_{i-1}\}$ degrades from component-wise level to subspace level.

### A2.2  ANOTHER PROPERTY

**Learning order matters.** Comparing both cases in Figure A1, they show the same identifiability considering all domains. However, we observe that in the top case, each new domain introduces a new changing variable, while in the bottom case, the domain order is reversed. Apparently, we can achieve subspace identifiability after learning each new domain in the top case, indicating that we

can progressively improve our understanding and representation of the underlying causal factors with the arrival of each new domain. However, we can only achieve subspace identifiability until learning all domains in the bottom case. This is in line with the current learning system, where we first learn subjects with fewer changes before moving on to subjects with more complexity.

## A3 PURSDO CODE

---

**Algorithm A1** Continual Nonlinear ICA

---

**Require:** Training data sequentially arriving $\{\mathbf{x}|\mathbf{u_1}, \ldots, \mathbf{x}|\mathbf{u_T}\}$
    Kaiming_init($\boldsymbol{\theta}$), $\mathcal{M}_t \leftarrow \{\}$ for all $t = 1, \ldots, T$
    **for** $\mathbf{u} = \mathbf{u_1}, \ldots, \mathbf{u_T}$ **do**:
        **for** $\{\mathbf{x_1}, \ldots, \mathbf{x_d}\}|\mathbf{u}$ **do**
            $\mathcal{M}_t \leftarrow \mathcal{M}_t \cup$ random select $\mathbf{x}$
            Calculate loss $\mathcal{L}(\boldsymbol{\theta})$ as equation 5
            $\mathbf{v} \leftarrow \nabla_\theta \mathcal{L}(\boldsymbol{\theta}, \mathbf{x})$
            $\mathbf{v}_k \leftarrow \nabla_\theta \mathcal{L}(\boldsymbol{\theta}, \mathcal{M}_k)$ for all $k < t$
            $\mathbf{v}' \leftarrow$ Solve quadratic programming as equation 7
            $\boldsymbol{\theta} \leftarrow \boldsymbol{\theta} - \alpha \mathbf{v}'$
    **Return** $\theta$

---

## A4 VISUALIZATION

To provide a more intuitive demonstration of the identifiability of changing variables and compare our method with joint training, we conducted an experiment in the following setting: with $n_s = 2, n = 4$, and $\mathbf{z}$ values generated from a Gaussian distribution across 15 domains. We stored models trained on subsets of the training data containing 3, 5, 7, and 9 domains, a part of the whole 15 domains respectively. The test set consisted of all 15 domains, and we used these models to sample corresponding $\hat{\mathbf{z}}$ values. These generated $\hat{\mathbf{z}}$ values were then compared to the ground truth values of $\mathbf{z}$ for evaluation.

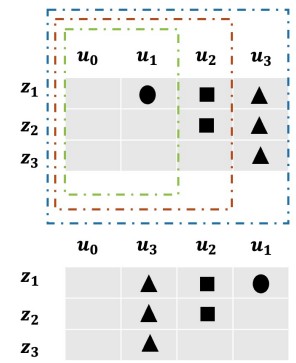

Specifically, we provide the scatter plot of true latent variables $\mathbf{z}$ with the estimated variables $\hat{\mathbf{z}}$ in Figure A2,A3,A4,A5 for both our methods and joint training. Figure A2,A3,A4,A5 corresponds to a different training set that includes 3, 5, 7, and 9 domains respectively. For each figure, $\hat{\mathbf{z}}_{s,i}$ represents the $i$th estimated changing variable, $\hat{\mathbf{z}}_{c,i}$ represents $i$th estimated invariant variable, $\mathbf{z}_{s,i}$ represents the $i$th true changing variable and $\mathbf{z}_{c,i}$ represents the $i$th true invariant variable.

Based on the experiment results, we observe a stronger linear correlation that appears for estimated changing variables with real changing

Figure A1: **A toy example with three variables and four domains.** $z_1$ changes in $\mathbf{u}_1, \mathbf{u}_2, \mathbf{u}_3$, $z_2$ changes in $\mathbf{u}_2, \mathbf{u}_3$, and $z_3$ changes in $\mathbf{u}_3$.

ones as more domains are included in the training process for both our method and joint training. That is, more domains will imply stronger identifiability, aligned with our expectations. Beyond that, our approach shows slightly inferior or even comparable performance compared to joint training, demonstrating its effectiveness.

## A5 EXPERIMENT DETAILS

### A5.1 DATA

We follow the data generation defined in Equation 1. Specifically, we discuss Gaussian cases where $\mathbf{z}_c \sim N(\mathbf{0}, \mathbf{I})$, $z_s \sim N(\mu_{\mathbf{u}}, \sigma_{\mathbf{u}}^2 \mathbf{I})$ for both $n_s = 2, n = 4$ and $n_s = 4, n = 8$. For each domain $\mathbf{u}$, the $\mu_{\mathbf{u}} \sim Uniform(-4, 4)$ and $\sigma_{\mathbf{u}}^2 \sim Uniform(0.01, 1)$.

Scatter Plot with 3 domains involves training: our method

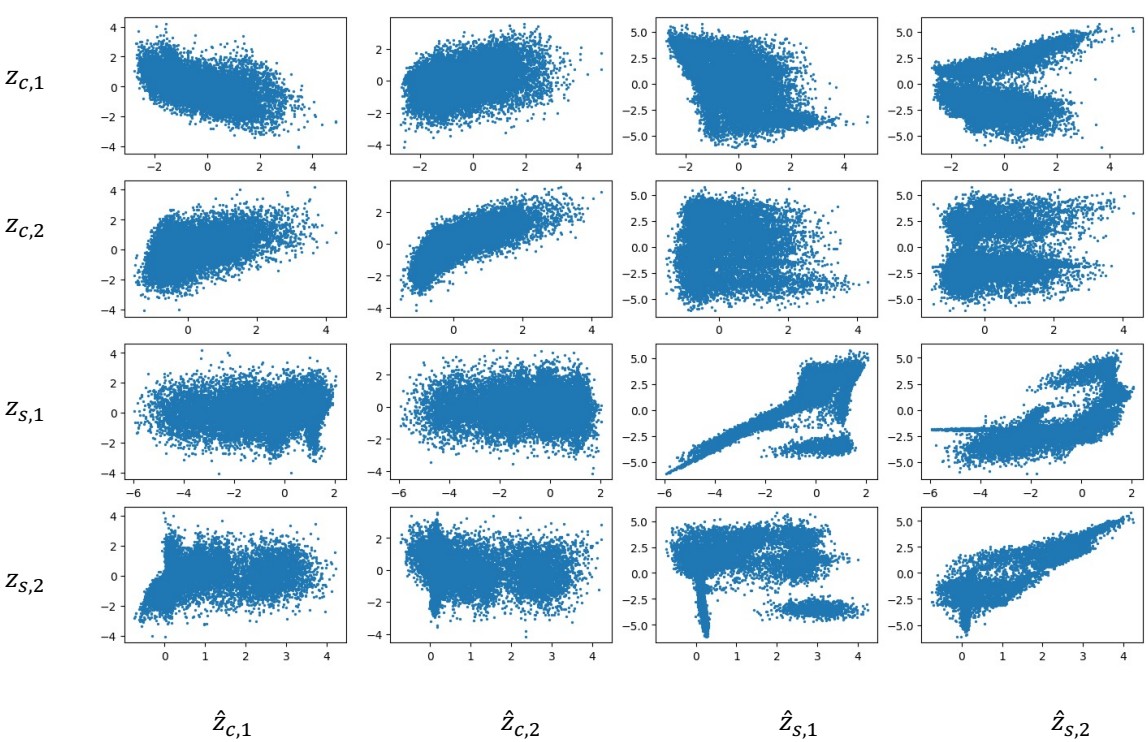

Scatter Plot with 3 domains involves training: joint training

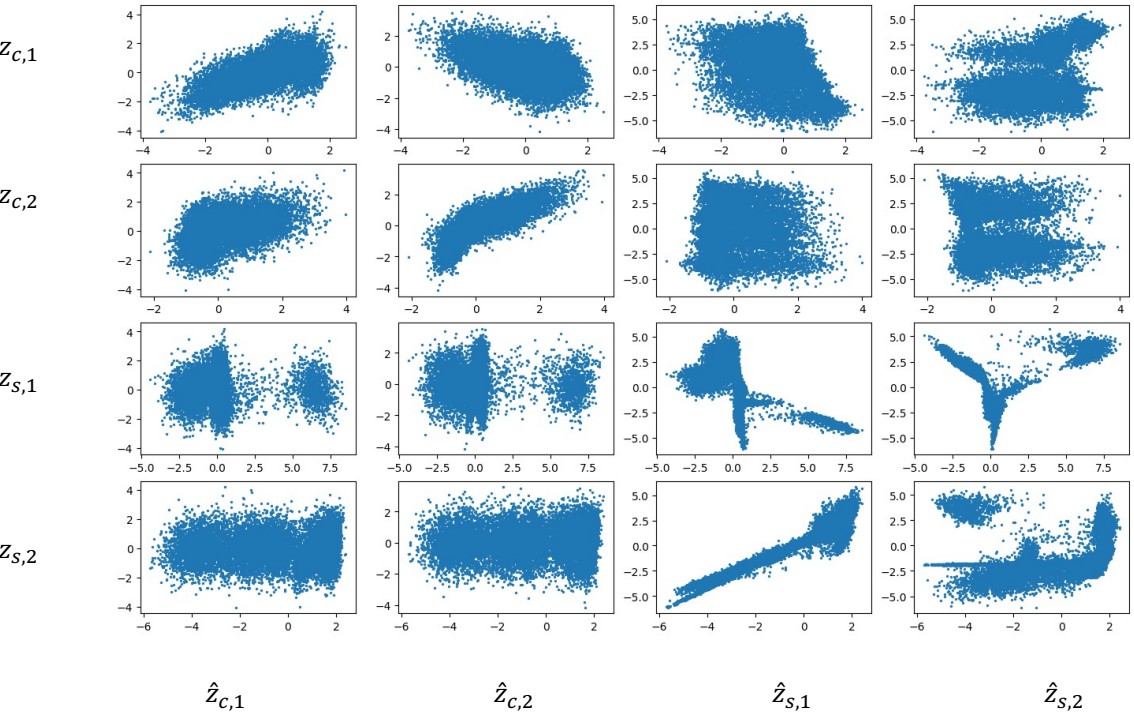

Figure A2: Visual comparison of our methods with joint training in setting that $\mathbf{z}$ are Gaussian, $n_s = 2, n = 4$. One should note that this shows the model evaluated over all 15 domains while 3 domains involve in training.

Scatter Plot with 5 domains involves training: our method

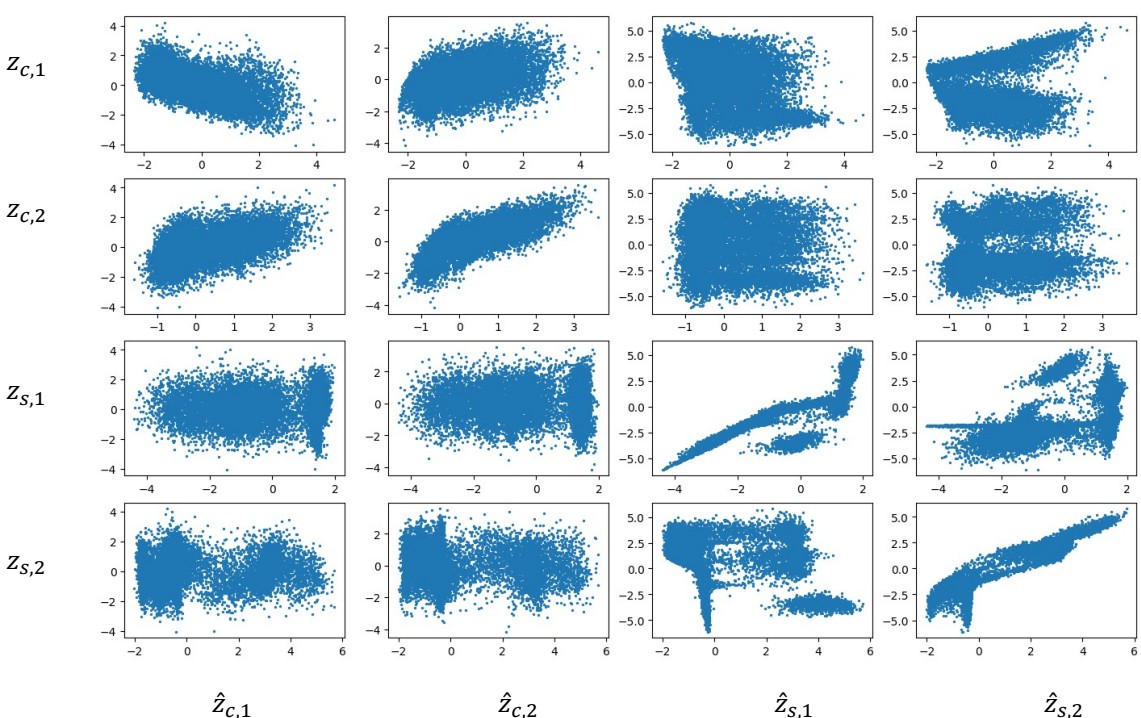

Scatter Plot with 5 domains involves training: joint training

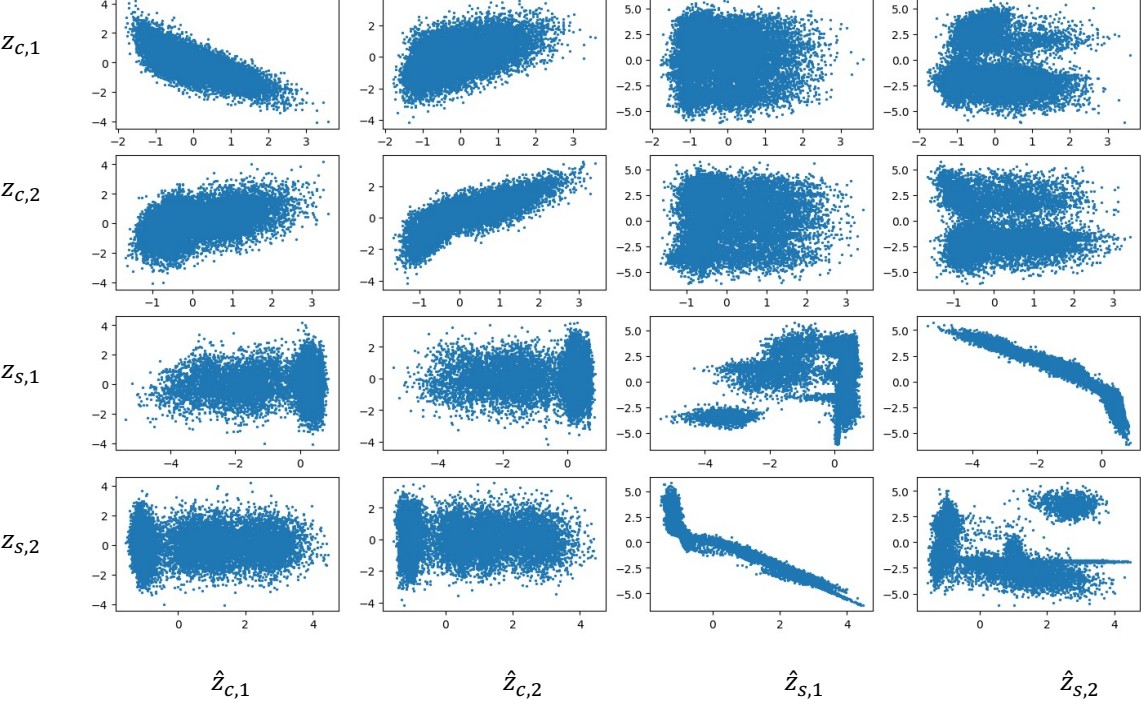

Figure A3: Visual comparison of our methods with joint training in setting that $\mathbf{z}$ are Gaussian, $n_s = 2, n = 4$. One should note that this shows the model evaluated over all 15 domains while 5 domains involve in training.

Scatter Plot with 7 domains involves training: our method

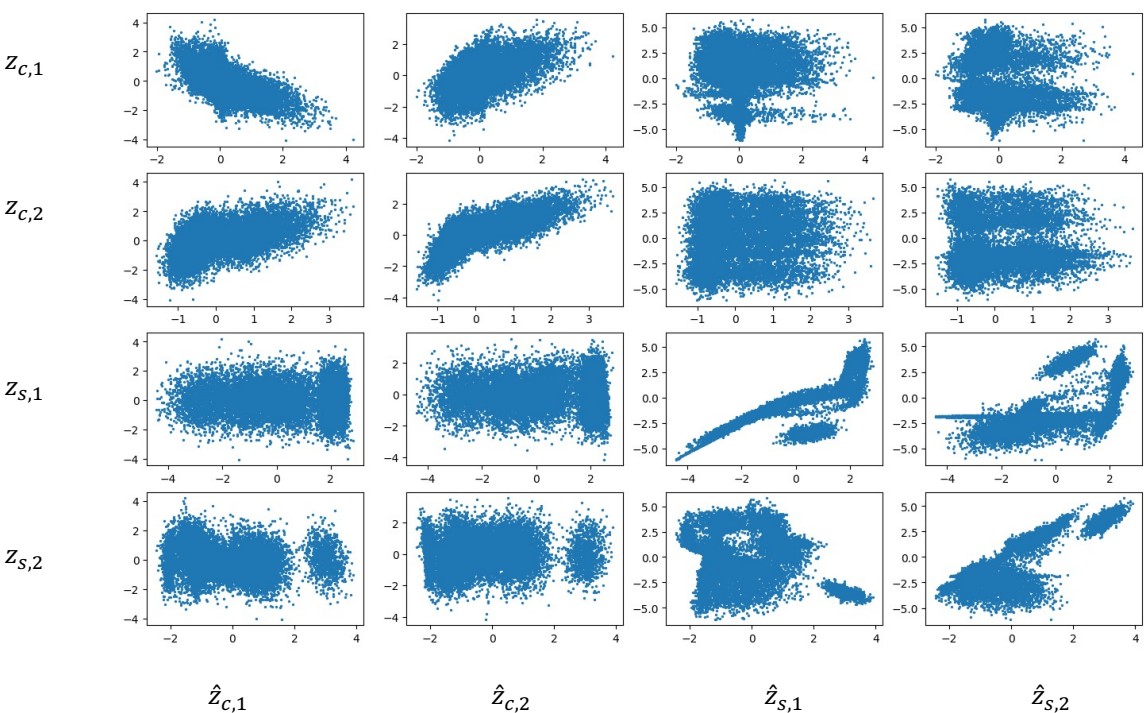

Scatter Plot with 7 domains involves training: joint training

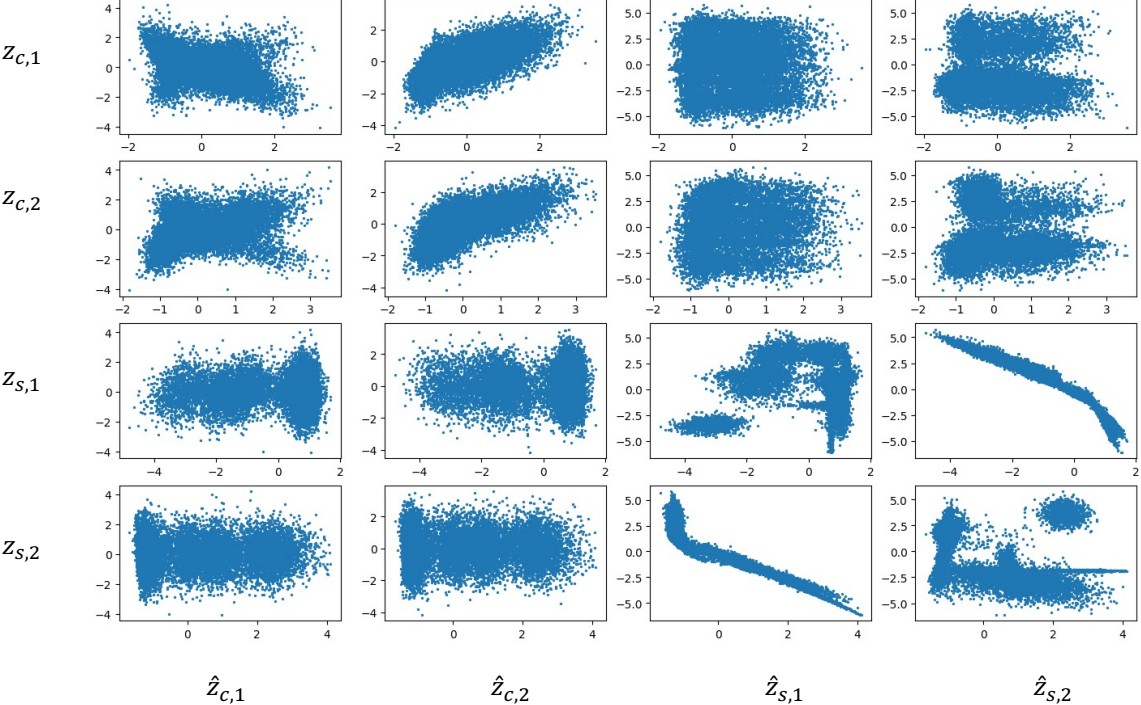

Figure A4: Visual comparison of our methods with joint training in setting that $\mathbf{z}$ are Gaussian, $n_s = 2, n = 4$. One should note that this shows the model evaluated over all 15 domains while 7 domains involve in training.

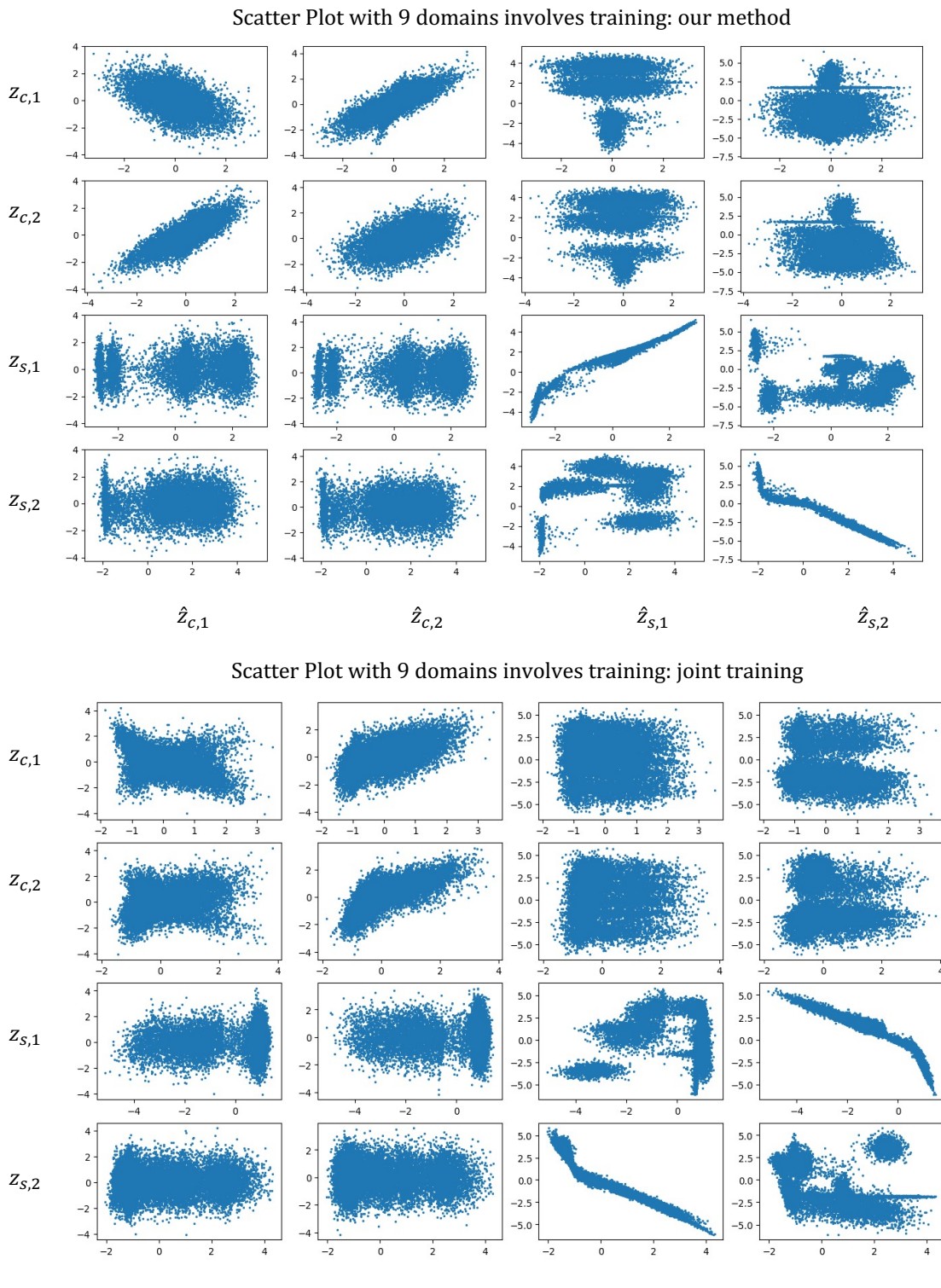

Figure A5: Visual comparison of our methods with joint training in setting that $\mathbf{z}$ are Gaussian, $n_s = 2, n = 4$. One should note that this shows the model evaluated over all 15 domains while 9 domains involve in training.

We also discuss the mixed Gaussian case or both $n_s = 2, n = 4$ and $n_s = 4, n = 8$ where $\mathbf{z}_s$ is the normalization of mixing of two Gaussian variables $N(\mathbf{0}, \mathbf{I})$ and $N(\mathbf{0.25}, \mathbf{I})$ with domain-specific modulation and translation. For $n_s = 2, n = 4$, each domain contains 10000 samples for training and 1000 samples for testing. For $n_s = 4, n = 8$, each domain contains 5000 samples for training and 1000 samples for testing. Specifically, we first mix those two Gaussian and do the normalization. After that, we modulate the normalized variable on every domain with a random variable sampled from $Uniform(0.01, 1)$. Then, we translate it with a random variable sampled from $Uniform(-4, 4)$.

## A5.2  MEAN CORRELATION COEFFICIENT

Mean correlation coefficient(MCC) is a standard metric for evaluating the recovery of latent factors in ICA literature. It averages the absolute value of the correlation coefficient between true changing variables with the estimated ones. As stated in our paper, the Lemma1 can only guarantee component-wise identifiability, leaving it unfair to directly calculate. e.g., $\mathbf{z} = \hat{\mathbf{z}}^2$ will give the correlation 0 (One should note this cannot happen in our case as $h(x) = x^2$ is not invertible, this is just an illustrative example).

We thus use MLP to remove the nonlinearity. Specifically, we first solve a linear sum assignment problem in polynomial time on the computed correlation matrix to pair the true variables with the estimated ones. We divide these pairs into a 50-50 ratio for the training set and the test set. For each pair, we train a MLP on the training set to do the regression to remove the nonlinearity and use the test set to calculate the correlation coefficient. We then average those correlation coefficients to get the final result.

## A5.3  NETWORK ARCHITECTURE AND HYPER-PARAMETERS

**Network Architecture.** We follow the design of Kong et al. (2022) and summarize our specific network architecture in Table A1, which includes VAE encoder and decoder. We use component-wise spline flows Durkan et al. (2019) to modulate the changing components $\hat{\mathbf{z}}_s$.

Table A1: Architecture details. BS: batch size, i_dim: input dimension, z_dim: latent dimension, s_dim: the dimension of latent changing variables, LeakyReLU: Leaky Rectified Linear Unit.

| Configuration | Description | Output |
|---|---|---|
| **1. MLP-Encoder** | Encoder for Synthetic Data | |
| Input | Observed data from one domain x\|u | BS × i_dim |
| Linear | 32 neurons | BS × 32 |
| Linear with Activation | 32 neurons, LeakyReLU | BS × 32 |
| Linear with Activation | 32 neurons, LeakyReLU | BS × 32 |
| Linear with Activation | 32 neurons, LeakyReLU | BS × 32 |
| Linear with Activation | 32 neurons, LeakyReLU | BS × 32 |
| Activation | LeakyReLU | BS × 32 |
| Linear | 32 neurons | BS × 2*z_dim |
| **2. MLP-Decoder** | Decoder for Synthetic Data | |
| Input: $\hat{\mathbf{z}}$ | Sampled latent variables | BS × z_dim |
| Linear | 32 neurons | BS × 32 |
| Linear with Activation | 32 neurons, LeakyReLU | BS × 32 |
| Linear with Activation | 32 neurons, LeakyReLU | BS × 32 |
| Linear with Activation | 32 neurons, LeakyReLU | BS × 32 |
| Linear with Activation | 32 neurons, LeakyReLU | BS × 32 |
| Activation | LeakyReLU | BS × 32 |
| Linear | 32 neurons | BS × i_dim |
| **6. Flow Model** | Flow Model to build relation between $\hat{\mathbf{z}}_s$ with $\hat{\tilde{\mathbf{z}}}_s$ | |
| Input | Sampled latent changing variable $\hat{\mathbf{z}}_s$ | BS × s_dim |
| Spline flow | 8 bins, 5 bound | BS × s_dim |

**Training hyper-paramters** We apply Adam to train our model with 50 epochs for every domain. We use a learning rate of 0.002 with batch size of 256. We set $\alpha$ and $\beta$ both 0.1 in calculating loss as Equation 5. For the memory size of $\mathcal{M}|\mathbf{u}$, we randomly select 256 samples of each domain. The negative slope in Leaky-Relu is set to be 0.2.

**Code** We include our code in the same zip file. For the reproduction detail, one can refer to the readme file.

