# OpenReview forum: "Continual Nonlinear ICA-Based Representation Learning"
_ICLR.cc/2024/Conference — Submitted to ICLR 2024_

### Official Review · Reviewer_k5LY · 2023-10-31

**Soundness:** 2 fair
**Presentation:** 2 fair
**Contribution:** 2 fair
**Rating:** 3
**Confidence:** 2

**Summary:**

This paper studies an interesting topic: causal representation learning. Recent works in nonlinear Independent Component Analysis (ICA) provide a promising causal representation learning framework by separating latent sources from observable nonlinear mixtures. This paper introduces a new approach that optimizes the model by satisfying two objectives: (1) reconstructing the observations within the
current domain, and (2) preserving the reconstruction capabilities for prior domains through gradient constraints. Experiments show that the proposed approach can achieve good performance.

**Strengths:**

1. This paper is well-written.
2. The research topic in this paper is interesting.

**Weaknesses:**

1. It is not clear how you address continual learning.
2. What is the actual form for the domain variable u?
3. This paper only considers a general continual learning setting, which relies on the task information. However, the proposed approach can not be used in task-free continual learning.
4. The theoretical framework is based on the existing work (Kong et al., 2022).
5. The number of baselines in the experiment is small and more continual learning experiments should be performed.
6. Since this paper employs the generative model. The lifelong generative modelling experiments should be provided.

**Questions:**

Please see the weakness section.

---

> ### Author Response · Authors · 2023-11-20
> **We are very grateful for your time, insightful comments, and encouragement**
>
> We are very grateful for your time, insightful comments, and encouragement. Below please see our point-by-point response.
>
> > It is not clear how you address continual learning.
>
> Thank you for your question. As detailed in Section 3.3, we address continual learning through pursuing two objectives: (1) the reconstruction of observations within the current domain and (2) the preservation of reconstruction capabilities for preceding domains. Specifically, we use GEM algorithm to enforce that the movement of network parameters learning a new domain should not result in an increased loss for the previous domains.
>
> > What is the actual form for the domain variable $u$?
>
> $u$ is an indicator of domain which can be represented with a vector or a scalar.
>
> > This paper only considers a general continual learning setting, which relies on the task information. However, the proposed approach can not be used in task-free continual learning.
>
> Thank you for your question and we would like to re-emphasize the difference between ours with traditional continual classification tasks. In the traditional continual classification task, each task is independent. The purpose of continual learning is to make the model remember each task as much as possible. In this situation, not knowing the identities of tasks will indeed affect testing. However, for our identifiable nonlinear ICA learned in sequentially arrived domains, we need multiple domains to make the model identifiable. Once we have enough domains, knowing the identity of the domain (task) or not is not important at all for the test phase. Regardless of which domain the tested data comes from, even a completely unseen domain, we can still recover meaningful latent variables (component-wise identifiable). Figure 5 (a) also validates this point as the model is always tested on all 15 domains while the training data increases gradually from 1 domain to 15 domains.
>
> For the training phase, it should be noted that our observations are generated from changing distribution through **fixed** transformations. Thus, it would be easy to capture the boundary of each domain for model training.
>
> > The theoretical framework is based on the existing work [1].
>
> Thanks for your comment. Compared with the framework from [1], we introduced the subspace identifiability for changing variables, which is essential to investigate the case where few domains are observed. Beyond that, we have demonstrated that, in the context of continual learning, the identifiability of certain variables becomes compromised with the introduction of new domains, as outlined in Proposition 2 and empirically shown in Figure 6.
>
> > The number of baselines in the experiment is small.
>
> The reason we have not conducted comparisons of our method with other nonlinear ICA approaches is due to the distinctive structure of our model. Specifically, our approach involves partitioning latent variables into two distinct categories: changing and invariant components. Thus, other nonlinear ICA frameworks cannot be directly compared as a baseline.
>
> > More continual learning experiments should be performed.
>
> We are currently conducting the experiment on image datasets and will report the results once we get them.
>
> > The lifelong generative modeling experiments should be provided.
>
> We appreciate your comment, but we would like to clarify our understanding of your question to provide a more accurate response. Section 4 of our paper is dedicated entirely to the lifelong generative modeling experiment. If you could provide more specific details or context regarding your query, it would greatly assist us in addressing your concerns or questions more effectively.
>
> [1]. Kong. et al. Partial Disentanglement for Domain Adaptation, 2022

---

### Official Review · Reviewer_iGtV · 2023-11-02

**Soundness:** 2 fair
**Presentation:** 1 poor
**Contribution:** 3 good
**Rating:** 3
**Confidence:** 4

**Summary:**

This paper studies identifiability of VAE models trained on a stream of different data domains. The ground truth generative model assumes that one part of the latents is domain-dependent, and another part domain-independent.

**Strengths:**

I really enjoyed reading the paper. I think the motivation is clear, the problem interesting and relevant. Some theoretical and algorithmic innovation is given, and the theory is validated by synthetic experiments (but see weaknesses below).

**Weaknesses:**

The biggest weakness by far is the empirical investigation. The theory, while interesting and relevant, seems too incremental to justify acceptance just based on the theoretical contribution --- more empirical validation and comparisons to prior work is needed. The main weaknesses are:

- There is no discussion of potentially competing ICA approaches in the literature, and no comparison to baseline algorithms. The evaluation is only with regards to the proposed setting, no external validation takes place. Especially the joint training setup is applicable to a variety of non-linear ICA methods, so a better empirical comparison would greatly enhance the positioning w.r.t prior work.
- It is a bit tricky to connect the sections in the appendix to the theorems/proposition of the paper. It would enhance readability if there are clear headings in the appendix, and/or the authors would re-state the results from the paper.
- In the derivation in A1 and the following sections, the arguments to the Jacobians, e.g. $J_h$, are dropped. I find it not always clear from the context to infer the arguments. I am especially wondering (but might be wrong) whether the Jacobian $J_h$ depends on $\mathbf u$, see my question below. In any case, stating the argument of $J_h$ would improve readability of the proof.
- between Eqs (21), (22), is it necessary to keep the $\mathbf 0$ argument? I find this more confusing/uncessary than helpful, but might overlook something.
- The method is purely validated on synthetic toy datasets. There's a wealth of e.g. image datasets available (dsprites and the like) to validate the approach on more complex distributions, without drastically increasing the number of latent factors. Such an exploration would improve the paper a lot, I would be happy to discuss a choice of datasets with the authors before running experiments. This could be especially interesting to "convert" an existing benchmark into the continual ICA setting.
- There are a lot of typos in the appendix and proofs, in general more care could be taken with spellchecking and typesetting. A common typesetting issue is missing spaces, or inconsistent upper/lowercasing (assumption / Assumption, etc). This should be fixed.
- A2, Proposition 2, typesettng errors ($<\le$ etc).

My current assessment is based on the current state of the paper, which can be improved in terms of clarity in the theory (esp. in the appendix) and the experimental results (comparisons to more baseline methods from the literature, scaling beyond synthetic data). I think with good execution and improvement along these dimensions, the current paper story and problem setting could easily get a 6 or even 8 in score, and I expect to re-adapt my evaluation during the rebuttal phase based on the the improvements made.

**Questions:**

- Figure 5: The error bars are quite large between baseline and joint --- did you run a test whether the improvements observed are signficant?
- Performance of the empirical validation is far from optimal, MCCs are substantially smaller than 1. What would it take to observe a result close to the "theoretical limit"? Have you considered how different components (number of latents, number of samples in the dataset, ...) influence the final result?
- In A1, does $J_h$ depend on $\mathbf u$?
- "the distribution estimate variable $\tilde z_j$ doen't change across all domains" -> Can you clarify why this is? That statement is not obvious to me in the context of the proof.
- "Similarly, $q_i \dots$ remainds the same for ...$ -> same concern, not obvious, maybe a ref is needed.
- Is there a reason why $\mathbf u$ is assumed to be a vector? for the purpose of the proof, isn't it sufficient to assume an integer value? (the function $f_u$ might still map it to a vector internally, I am just not sure why that assumption is needed).

---

> ### Author Response · Authors · 2023-11-20
> **We are very grateful for your time, insightful comments, and encouragement - Part 1**
>
> We sincerely thank the reviewer for the time dedicated to reviewing our paper, the constructive suggestions, and encouraging feedback. Please find the response to your comments and questions below.
>
> > In the derivation in A1 and the following sections, the arguments to the Jacobians, e.g. $J_h$, are dropped. Is $u$ related to $J_h$?
>
> Thank you for your question. $u$ is not related to $J_h$. The transformation $h = g^{-1} \circ g$ describes the relationship between the true latents $z$ and the estimated variables $\hat{z}$, which will not be influenced by the change of domains $u$. For different $u$, Jocabian $J_h$ is the same and the existence of $u$ is to help us to remove the intractable term $J_h$. We will make this clear in the revised manuscript.
>
> Let us summarize the overall logic of the proof: we can easily construct the relationship between estimated variables $z$ and true latents $\hat{z}$ like Eq(16) as both $g$ and $\hat{g}$ is invertible. The challenge is that we do not know about $J_h$ (once we know, everything is solved), while we want to investigate the relationship between $\hat{z}$ with $z_s$. To solve the untractable term $J_h$, we introduce the domain $u$ and produce multiple equations like Eq~(17). Then, we can remove the intractable Jocabian term by taking the difference for every equation with the equation where $u=u_0$. This is why the $u$ is not related to $J_h$.
>
> >"the distribution estimate $\hat{z}_j$ variable doesn't change across all domains" and "Similarly, q_i remains the same for"
>
> The reason that why  $\hat{z}_j$ doesn't change across all domains is as follows: the $\hat{z}_j$ is defined to be the estimated invariant variable that will not be affected by domain $u$, for $j \in [n_s+1, \dots, n]$. Similarly, $q_i(z_i, u)$ for $i \in [n_s+1, \dots, n]$ represents the log density of groundtruth invariant variables, and thus it also does not change across all domains.
>
> > Is there a reason why $ u$ is assumed to be a vector? for the purpose of the proof, isn't it sufficient to assume an integer value?
>
> Thanks for your insightful observation. We totally agree with you that it is sufficient to assume an integer value for $u$ for the purpose of the proof. In our framework, $u$ serves as an indicator to denote different domains, and it can indeed be represented either as a vector or a scalar. We opted for a vector representation to maintain generality (when dimension reduces to one, a vector degrades to a scalar). We will update the manuscript to include this discussion.
>
> > between Eqs (21), (22), is it necessary to keep the argument $0$ ?
>
> Thanks for your suggestion which helps improve the presentation and lighten the notations. We have updated the manuscript to remove the argument $0$.
>
> > The error bars are quite large between baseline and joint --- did you run a test whether the improvements observed are significant?
>
> We appreciate your insightful observation. All experiments shown in Figure~5 are run with seeds set at 1,2,3. The error bars of baselines are indeed large--a possible reason is that the performance of the baseline is heavily influenced by the last domain, giving rise to the large variance. In contrast, the error bars or variances of our proposed method and the joint approach are much smaller, which further indicates the importance of a continual learning approach in this setting. We will include this observation and discussion in the experiment section.
>
> > Performance of the empirical validation is far from optimal, MCCs are substantially smaller than 1. What would it take to observe a result close to the "theoretical limit"?
>
> Thanks for your question. A possible reason is that the estimation procedure involves neural networks, and the overall optimization problem is nonconvex, as is typical for methods involving deep learning. Thus, the estimation process, even for the joint training approach, may never achieve the global optimum. Another reason is that we divide the latent variables into invariant and changing parts, which may be more challenging than regular nonlinear ICA where all latents are changing. In fact, similar observations have been reported in [1].
>
> > Have you considered how different components (number of latents, number of samples in the dataset, ...) influence the final result?
>
> Thanks for your question. Kindly refer to Figure 4 in the paper which shows the influence of different components ($n_s=4,n=8$ and $n_s=2, n=4$) on the final MCC results. We also conducted an ablation study to investigate the influence of $\hat{n}_s$ as shown in Table~1.
>
> For different number of samples, in light of your suggestion, we conducted additional experiment results. For the case of $n_s=4,n=8$, we conducted training using 5000 samples and testing with 1000 samples for each domain. For the scenario of $n_s=2, n=4$, we conducted training using 10000 samples and testing with 1000 samples for each domain. We will add these details in the appendix of the final version.

---

> > ### Author Response · Authors · 2023-11-20
> > **We are very grateful for your time, insightful comments, and encouragement - Part 2**
> >
> > > There is no discussion of potentially competing ICA approaches in the literature, and no comparison to baseline algorithms.
> >
> > Thanks for pointing this out. The setting we considered is slightly different from the regular nonlinear ICA setting; that is, our setting divide the latent variables into changing and the invariant parts, which may be considerably more general. (In this case, regular nonlinear ICA considers all latent variables to be "changing".) For this setting involving changing and invariant parts, the proposed method by [1] is the state-of-the-art method which we adopt and compare to, and we are not aware of any other methods that consider the same setting. If there is any other specific method considering this setting that the reviewer thinks should be compared, we would greatly appreciate if you could kindly let us know.
> >
> > > The method is purely validated on synthetic toy datasets.
> >
> > Thanks for your suggestion. In light of your comment, we are currently conducting the experiment on image datasets and will report the results once we get them.
> >
> > [1]. Kong. et al. Partial Disentanglement for Domain Adaptation, 2022

---

> ### Comment · Reviewer_iGtV · 2023-11-20
> **Quick response to rebuttal**
>
> Dear authors, thanks a lot for your comments. I will go through your comments regarding the theory in more detail (from skimming them, they seem to address many of my open Qs).
>
> I wanted to quickly chime in and confirm that additional (convincing) empirical results would be the main reason for me to change my evaluation of the paper. I appreciate that you decided to run additional experiments, and am happy to engage in further discussion once first results are in.

---

> > ### Comment · Reviewer_iGtV · 2023-12-02
> >
> > I wanted to thank the authors again for their rebuttal. However, the biggest weakness, limited empirical evaluation and comparison to prior work, was not addressed during the discussion phase, which is why I will keep my current score. I think the ICA setting and method in this paper is very interesting though, and this could become a nice contribution if the mentioned weaknesses are worked out and more empirical results are given in an upcoming revision.

---

### Official Review · Reviewer_r8sE · 2023-11-06

**Soundness:** 3 good
**Presentation:** 2 fair
**Contribution:** 2 fair
**Rating:** 3
**Confidence:** 4

**Summary:**

This paper considers the identifiability of nonlinear ICA in the continual learning setting i.e. under a changing and partially changing domains. The identifiability theorems show that under a sufficiently large number of domains with significant changes, the latent components can be identified up to component-wise nonlinearity. For a lower number of domains, subspace identifiability is still guaranteed as long as there are more domains than changing variables. A learning algorithm based on VAE and GEM (method for continual learning) is introduced and its performance (on identifiability) is presented on simulated data.

**Strengths:**

The paper's idea to identify nonlinear ICA in terms of sequential learning is novel. The authors nicely show how as the number of domains increases, the identifiability improves which is an interesting, albeit expected, result.

**Weaknesses:**

There are some major weaknesses and questions.

**1. Contribution is not clear since relevant previous work is ignored:**:

a). You claim that *"[Nonlinear ICA] still relies on observing sufficient domains simultaneously..."*. -- Not true necessarily: for example [1], shows the identifiability of hidden Markov nonlinear ICA. As is well known, HMMs can be learned sequentially / in online fashion with the latent states analogous to different domains. Authors have not covered relation to this work.

b). Related to above: [1] and [2] give a broad identifiability theorem that applies to situation with a changing latent variable that can be interpreted as the changing domain that alters the probability distributions i.e. it seems to already to cover mostly what is considered in the identifiability theorems here, and provides stronger results in a more general setting. It is therefore important to contrast to this work and explain what is novel here. Note that I understand that the sequential learning is different, but here I am only talking about the identifiability theorem. Other important and similar work is also ignored and need to be discussed such as [3]-[6]. Note also that most of these works *do not* assume auxiliary variables.

c). The paper is frames itself as "Causal Representation Learning" but it seems like it's much more related to nonlinear ICA and doesn't learn latent causal relationships -- see [6] and [7]. I recommend the authors to reconsider the use of this term.

**2. Experimental evaluation is lacking**

a). The author's only evaluate their model on synthetic data, no real data is used and the importance of this method to practicable applications is not clear

b.) Baseline does not include any of relevant previous works e.g. [1] could be reasonable adapted to this data


**3. Several issues on theory and estimation algorithm**

a.) The identifiability theorem appears not to consider observations noise -- yet the estimation method VAE clearly includes that. This mismatch and its impact on identifiability can be significant (Theorems 1, 2 in [2]) but this is ignored here

b.) The identifiability theorem does not appear to consider dimension reduction into the latent space, which is in practice very important. Without that you are usually stuck with high dimensional latent space -- and the theorems presented here are unlikely to hold with high dimensional latent variables making it unclear how useful the work is in practice.


**Other issues:**

a). You use "component-wise identifiability" but really this is inaccurate, you do not identify the elements component-wise, rather you identify them up to component-wise *nonlinearity*. This needs to be fixed in order to make sure the reader doesn't misunderstand and think that you can exactly identify each components.

b). "Importantly, the guarantee of identifiability persists even when incoming domains do not introduce substantial changes for partial variables". Please clarify what is meant by partial variables -- the term has not been defined by this point.

c). It is not explained clearly enough that the conditions in Theorems 1 and 2 are *sufficient* conditions, not *necessary* and I feel there is some confusing writing related to this. You for example say that "However, when considering domains u0, u1, u2, the component-wise identifiability for z1 disappears, and instead, we can only achieve subspace identifiability for both z1 and z2." This is not necessarily true -- according to your own theorems you can only *guarantee* subspace identifiability (sufficiency) but since theorem 1 does not give *necessary conditions* you can not say that "the component-wise identifiability for z1 disappears," it could still be component-wise identified,

misc.:
- the z in Figure 2 should not be bold if I'm correct
- poor grammar: "will contribute to both domains and we remain the direction.", "where no loss increment for previous domains."
- Figure 5 coming before Figure 4 is extremely confusing
- "We evaluate the efficacy of our proposed approach by comparing it against the same model trained on sequentially arriving domains and multiple domains simultaneously, referred to as the baseline and theoretical upper bound by the continual learning community." Could you please clearly define "joint" and "baseline" so that the reader can understand them.
- I dont see "Table 1" mentioned anywhere in the text, making it hard to understand

[1]. Hälvä and Hyvärinen, Hidden Markov Nonlinear ICA: Unsupervised Learning from Nonstationary Time Series, 2020

[2]. Hälvä et al. Disentangling Identifiable Features from Noisy Data with Structured Nonlinear ICA, 2021

[3]. Klindt et al. Towards Nonlinear Disentanglement in Natural Data with Temporal Sparse Coding, 2020

[4]. Gresele et al. The Incomplete Rosetta Stone Problem: Identifiability Results for Multi-View Nonlinear ICA, 2019

[5]. Gresele et al. Independent mechanism analysis, a new concept?, 2021

[6]. Hyvärinen et al. Identifiability of latent-variable and structural-equation models: from linear to nonlinear, 2023

[7]. Schölkopf et al. Towards Causal Representation Learning, 2021

**Questions:**

Q1. You write that "Thus, a fully identifiable nonlinear ICA needs to satisfy at least two requirements: the ability to reconstruct the observation and the complete  consistency with the true generating process. Unfortunately, current research is far from achieving this level of identifiability." Could you please explain what you mean by "far from achieving this level"?

Q2. Could you please explain how realistic the assumption in equation (3) is? After all, the method here is somewhat heuristic so we can not expect MLE-style guarantees

Q3. Am I correct understanding that this model does not allow dimension reduction into latent space?

Q4. The two paragraphs relating to Figure 2 are very hard to understand. In fact, the paragraph break between "Given our subspace identifiability theory, z1 can achieve subspace identifiability." and "As there is no change in the other variables in those two domains, this subspace identifiability is equal to competent-wise identifiability." doesn't seem to make sense. Why is there a paragraph break here? Further why is there $u_3$ if it's not mentioned in the text?

Q5. You write "Contrasted with the traditional joint learning setting, where the data of all domains are overwhelmed, the continual learning setting offers a unique advantage. It allows for achieving and maintaining original identifiability, effectively insulating it from the potential "noise" introduced by newly arriving domains." I think this is really a key, and very interesting if true, but as far as I understand, there is no theoretical way of proving this exactly? Am I correct in understanding this is more what you believe Theorem 1 implies?

Q6. "Assuming the function is invertible, we employ a flow model to obtain the high-level variable"-- Supposedly this high-level variable is not identifiable however? What is the impact of its unidentifiability?

Q7. What does the apostrophe in eq (6) mean?

Q8. I am bit confused by the lack of detail in the experiments section -- how is the experiment in Figure 5 different from the top-right one in Figure 4.

Q9. "We evaluate the efficacy of our proposed approach by comparing it against the same model trained on sequentially arriving domains and multiple domains simultaneously, referred to as the baseline and theoretical upper bound by the continual learning community." Could you please clearly define "joint" and "baseline" so that the reader can understand them.

Q10. "model achieves the component-wise identifiability and the extra domains (from 9 to 15) do not provide further improvement." How can you be sure it really is component-wise identifiable? Are all the components around similarly identified or is there a lot of variance?

Q11. why is there so much variance in the other methods in Figure 5b.)?

---

> ### Author Response · Authors · 2023-11-20
> **We sincerely thank the reviewer for the time dedicated to reviewing our paper, the constructive suggestions, and encouraging feedback - Part 1**
>
> We sincerely thank the reviewer for the time dedicated to reviewing our paper, the constructive suggestions, and encouraging feedback. Please find the response to your comments and questions below.
>
> > You claim that "[Nonlinear ICA] still relies on observing sufficient domains simultaneously...". -- Not true necessarily: for example [1], shows the identifiability of hidden Markov nonlinear ICA. As is well known, HMMs can be learned sequentially / in online fashion with the latent states analogous to different domains.
>
> Thank you for your insightful observation. It is indeed correct that Hidden Markov Models (HMMs) can be learned sequentially. However, as delineated in reference [1], the identifiability of the model is contingent upon the concurrent observation of a sufficient number of latent states or domains. On one hand, the validity of Equation (22) is compromised in scenarios where the model experiences 'forgetting', particularly when only a single latent state is observed at any given time. This scenario presents a stark contrast to the framework and contributions of our study, which focuses on a genuinely continual learning approach. On the other hand, to preserve the identifiability of the model in a sequential learning context (as proposed in [1]), where only one latent state is observed at a time, the algorithm we have used in our paper could be adapted as it only focuses the optimization procedure.
>
> > it seems to already to cover mostly what is considered in the identifiability theorems here, and provides stronger results in a more general setting. It is therefore important to contrast to this work and explain what is novel here.
>
> Thank you for your feedback. We agree that [1] and [2] are highly relevant, which we will discuss further in the revised manuscript. At the same time, we would also like to point out the differences with our setting (and therefore one is not "stronger" than another):
>
> (1) While [2] is predicated on exploiting the temporal structure of latent variables, our paper delves into a scenario where the latent variables are independently and identically distributed (i.i.d) within each domain. This distinction is crucial as the reliance on the use of temporal structures in [1] and [2] can indeed facilitate stronger identifiability results.
>
> (2) Our papers separate the latent variables into invariant and changing part, which is practical in real-world scenarios. We thoroughly investigated the conditions under which these two groups can be segregated (requiring at least $n_s+1$ domains) and also provided the sufficient condition that each changing variable is identified up to a nonlinear transformation ($2n_s+1$ domains). This exploration into the separability and identification of latent variables underscores the novelty and depth of our study.
>
> (3) In contrast to [1], which presupposes the conditional distribution of latent variables within the exponential family framework, our work does not make such an assumption.
>
> > Other important and similar work is also ignored and need to be discussed such as [3]-[6]. Note also that most of these works do not assume auxiliary variables.
>
> Thank you for sharing the references. We have provided a brief discussion on related works that do not assume auxiliary variables; see the third paragraph of Section 1. Given your suggestion, we will cite and discuss [3]-[6] in the revised version of our paper. We will also modify the sentence "it still relies on observing sufficient domains simultaneously" to "many of them still rely on observing sufficient domains simultaneously", to avoid any possible confusion.
>
> > The paper frames itself as "Causal Representation Learning" but it seems like it's much more related to nonlinear ICA and doesn't learn latent causal relationships -- see [6] and [7]. I recommend the authors to reconsider the use of this term.
>
> Thanks for your suggestion. We have reconsidered the use of the term and will use "continual identifiable representation learning" instead.

---

> > ### Author Response · Authors · 2023-11-20
> > **We sincerely thank the reviewer for the time dedicated to reviewing our paper, the constructive suggestions, and encouraging feedback - Part 2**
> >
> > >  The author's only evaluate their model on synthetic data, no real data is used and the importance of this method to practicable applications is not clear
> >
> > Thanks for your suggestion. We are currently conducting the experiment on image datasets and will report the results once we get them.
> >
> > > Baseline does not include any of the relevant previous works
> >
> > Thanks for pointing this out. The setting we considered is slightly different from the regular nonlinear ICA setting; that is, our setting divides the latent variables into changing and invariant parts, which may be considerably more general. (In this case, regular nonlinear ICA considers all latent variables to be "changing".) For this setting involving changing and invariant parts, the proposed method by [8] is the state-of-the-art method which we adopt and compare to, and we are not aware of any other methods that consider the same setting. If there is any other specific method considering this setting that the reviewer thinks should be compared, we would greatly appreciate if you could kindly let us know.
> >
> > > The identifiability theorem appears not to consider observation noise -- yet the estimation method VAE clearly includes that. This mismatch and its impact on identifiability can be significant (Theorems 1, 2 in [2]) but this is ignored here.
> >
> > Thank you for your valuable feedback. We acknowledge that our current identifiability theory does not incorporate observation noise and does not facilitate dimension reduction. However, it is important to emphasize that the crux of our research is the identification of nonlinear ICA in a continual learning context. This aspect, to the best of our knowledge, has not been well explored in existing literature.
> >
> > Our work revolves around the concept that the identifiability of nonlinear ICA, reliant on multiple domains. This principle holds true even in scenarios where auxiliary variables may not be directly observable; the essence of making the model identifiable remains rooted in utilizing the distribution change of latent variables.
> >
> > The primary objective of this paper is to elucidate a methodology for addressing scenarios where these distributional changes in latent variables cannot be observed concurrently. This approach mirrors the human cognitive process of sequentially observing and refining understanding to discern true latent factors.
> >
> > We also recognize the significance of considerations regarding noise and dimension reduction. These factors are indeed crucial in the broader context of model development and application. There are multiple possible solutions to solve this problem. For example, we can use PCA to first reduce the dimension of the observation and then apply our nonlinear ICA framework. We can also follow the work of [7] to propose an explicit noise model. Therefore, we plan to extend our research to include these aspects in our future work.
> >
> > > You use "component-wise identifiability" but really this is inaccurate
> >
> > Thanks for your suggestion. We will add a footnote to avoid such kind of confusion.
> >
> > > Please clarify what is meant by partial variables -- the term has not been defined by this point
> >
> > Sorry for the confusion. What we want to express here is even when a new domain comes in, there is a part of changing variables that don't undergo any distribution change. Kindly refer to Section 3.2.1 for the details.
> >
> > >  It is not explained clearly enough that the conditions in Theorems 1 and 2 are sufficient conditions, not necessary conditions.
> >
> > Sorry for the confusion. We will refine our paper and change the expression to "the component-wise identifiability cannot be guaranteed anymore".
> >
> > >  Could you please explain how realistic the assumption in equation (3) is?
> >
> > Equation (3) can be effectively achieved with a generative model.
> >
> > > You write "Contrasted with the traditional joint learning setting, where the data of all domains are overwhelmed, the continual learning setting offers a unique advantage. It allows for achieving and maintaining original identifiability, effectively insulating it from the potential "noise" introduced by newly arriving domains." I think this is really a key, and very interesting if true, but as far as I understand, there is no theoretical way of proving this exactly. Am I correct in understanding this is more what you believe Theorem 1 implies?
> >
> > Thanks for your question. It's true that we can't prove it theoretically as Theorem1 and Lemma1 state that the identifiability of latent variables only relates to the distribution of those latent variables itself as long as Equation(3) is satisfied. In other words,
> > the way of learning won't affect the theoretical identifiability. However, we show it empirically in Figure(6). It's more like a property when combine Theorem1 and Lemma1 together in the context of continul learning.

---

> > > ### Author Response · Authors · 2023-11-20
> > > **We sincerely thank the reviewer for the time dedicated to reviewing our paper, the constructive suggestions, and encouraging feedback - Part 3**
> > >
> > > >  "Assuming the function is invertible, we employ a flow model to obtain the high-level variable"-- Supposedly this high-level variable is not identifiable however? What is the impact of its unidentifiability?
> > >
> > > Thanks for your question. The existence of high-level variables is to help the experiment implementation. We believe the changing variables $\mathbf{z}_s$ is influenced by the domain $\mathbf{u}$. However, as the domain $\mathbf{u}$ is just an indicator, it's hard to produce the distribution of $\mathbf{z}_s$ from $\mathbf{z}_s$ directly. Thus, we introduce the high-level invariance $\mathbf{\tilde{z}}_s$ to help such implementation. The high-level invariance $\mathbf{\tilde{z}}_s$ is not identifiable but it's not important as we only care about the changing variables $\mathbf{z}_s$.
> > >
> > > > What does the apostrophe in eq (6) mean?
> > >
> > > It's just a symbol to distinguish the projected gradient from the original gradient.
> > >
> > > > How is the experiment in Figure 5 different from the top-right one in Figure 4.
> > >
> > > For Figure 5(a), the model is trained on increasing domains and tested on fixed all 15 domains. For the top-right one in Figure 4, the number of training and the number of testing domains are equated.
> > >
> > > > Could you please clearly define "joint" and "baseline" so that the reader can understand them.
> > >
> > > "Joint" is the regular model training where data from all domains can be simultaneously observed. "Baseline" refers to the case model sequentially trained on each domain without any modification and refinement to the training process.
> > >
> > > > Why is there so much variance in the other methods in Figure 5b.
> > >
> > > Thanks for your question. The primary observation from our analysis indicates that the most significant variance occurs in the baseline scenario, as represented by the green curve in our results. In this particular case, it's noteworthy that the performance of the model is heavily influenced by the last domain observed.
> > >
> > > > The two paragraphs relating to Figure 2 are very hard to understand.
> > >
> > > Thanks for your suggestion. We have refined the expression in our paper.
> > >
> > > > How can you be sure it really is component-wise identifiable? Are all the components around similarly identified or is there a lot of variance?
> > >
> > > Thanks for pointing this out. We will modify the sentence to "We observe that the MCC reaches a performance plateau at $9$ domains, and the extra domains (from 9 to 15) do not provide further improvement. This appears to be consistent with the identifiability theory that $2n_s+1=9$ domains are needed for identifiability."
> > >
> > > >  You write that "Thus, a fully identifiable nonlinear ICA needs to satisfy at least two requirements: the ability to reconstruct the observation and the complete consistency with the true generating process. Unfortunately, current research is far from achieving this level of identifiability." Could you please explain what you mean by "far from achieving this level"?
> > >
> > > Thanks for the question. We intend to say "current research cannot achieve this level of identifiability without further assumptions that are considerably restrictive". We will modify this sentence in the revised manuscript.
> > >
> > > [1]. Hälvä and Hyvärinen, Hidden Markov Nonlinear ICA: Unsupervised Learning from Nonstationary Time Series, 2020
> > >
> > > [2]. Hälvä et al. Disentangling Identifiable Features from Noisy Data with Structured Nonlinear ICA, 2021
> > >
> > > [3]. Klindt et al. Towards Nonlinear Disentanglement in Natural Data with Temporal Sparse Coding, 2020
> > >
> > > [4]. Gresele et al. The Incomplete Rosetta Stone Problem: Identifiability Results for Multi-View Nonlinear ICA, 2019
> > >
> > > [5]. Gresele et al. Independent mechanism analysis, a new concept?, 2021
> > >
> > > [6]. Hyvärinen et al. Identifiability of latent-variable and structural-equation models: from linear to nonlinear, 2023
> > >
> > > [7]. Khemakhem et al. Variational Autoencoders and Nonlinear ICA: A Unifying Framework, 2020
> > >
> > > [8]. Kong. et al. Partial Disentanglement for Domain Adaptation, 2022

---

> > > > ### Comment · Reviewer_r8sE · 2023-11-22
> > > >
> > > > I thank the reviewers for their rebuttal and it has cleared up several things but not the big problems -- thus I will hold my score as it is. I think there are currently too many issues with the paper.
> > > >
> > > > On the theoretical side, I am not convinced that this idea of "new domains" brings anything substantial new e.g. with respect to earlier work. In partcular as response to the author comments:
> > > >
> > > > >(1) While [2] is predicated on exploiting the temporal structure of latent variables, our paper delves into a scenario where the latent variables are independently and identically distributed (i.i.d) within each domain. This distinction is crucial as the reliance on the use of temporal structures in [1] and [2] can indeed facilitate stronger identifiability results.
> > > >
> > > > To above: actually, there is nothing to prevent this type of situation in [2] and it's in fact whathappens in [1].
> > > >
> > > > >(2) Our papers separate the latent variables into invariant and changing part, which is practical in real-world scenarios. We thoroughly investigated the conditions under which these two groups can be segregated (requiring at least domains) and also provided the sufficient condition that each changing variable is identified up to a nonlinear transformation ( domains). This exploration into the separability and identification of latent variables underscores the novelty and depth of our study.
> > > > >(3) In contrast to [1], which presupposes the conditional distribution of latent variables within the exponential family framework, our work does not make such an assumption.
> > > >
> > > > To above: I don't see why this idea isn't immediately from previous works. e.g. [1] we can assume that the domains are such that only some of the components distribution changes and we would have the same result as far as I can tell? As you say [1] does assume exponential family, which may be restrictive but [2] for example does not and seems to also naturally include the identifiability result of this type of changing and constant variables with domain changes.
> > > >
> > > > >Thank you for your insightful observation. It is indeed correct that Hidden Markov Models (HMMs) can be learned sequentially. However, as delineated in reference [1], the identifiability of the model is contingent upon the concurrent observation of a sufficient number of latent states or domains
> > > >
> > > > To above: I don't agree with this. Theoretical identifiability is a property of the model. How it is estimated is a separated a concept. In practice of course the two sides interact and may produce different results. But the identifiability theory of the HMM-NICA model ought to hold regardless of what type of algorithm is used.
> > > >
> > > > Finally, the experimentation is lacking as described and I propose the authors take in feedback from all the reviewers to improve their work. I would like to see for example how models from [1] or [2], after being adapted to this scenario, perform because I can't see them being different currently.

---

### Meta-Review · Area_Chair_VQ1T · 2023-12-13

**Metareview:**

This paper investigates the identifiability of nonlinear ICA in an online or continual learning setting. The reviewers identified some interesting new results in the paper, but in its current form it is not yet ready for publication at ICLR; multiple reviewers identified the need for stronger empirical testing and a comparison (both theoretical and empirical) with some existing prior work.

**Justification For Why Not Higher Score:**

All reviewers had concerns regarding experiments and references to prior work.

**Justification For Why Not Lower Score:**

N/A

---

### Decision · Program_Chairs · 2024-01-16

Reject